# Identification of functionally distinct macrophage subpopulations in *Drosophila*

Jonathon Alexis Coates[1], Elliot Brooks[2], Amy Louise Brittle[2], Emma Louise Armitage[2], Martin Peter Zeidler[1], Iwan Robert Evans[2]*

[1]Department of Biomedical Science and the Bateson Centre, University of Sheffield, Sheffield, United Kingdom; [2]Department of Infection, Immunity and Cardiovascular Disease and the Bateson Centre, University of Sheffield, Sheffield, United Kingdom

**Abstract** Vertebrate macrophages are a highly heterogeneous cell population, but while *Drosophila* blood is dominated by a macrophage-like lineage (plasmatocytes), until very recently these cells were considered to represent a homogeneous population. Here, we present our identification of enhancer elements labelling plasmatocyte subpopulations, which vary in abundance across development. These subpopulations exhibit functional differences compared to the overall population, including more potent injury responses and differential localisation and dynamics in pupae and adults. Our enhancer analysis identified candidate genes regulating plasmatocyte behaviour: pan-plasmatocyte expression of one such gene (*Calnexin14D*) improves wound responses, causing the overall population to resemble more closely the subpopulation marked by the *Calnexin14D*-associated enhancer. Finally, we show that exposure to increased levels of apoptotic cell death modulates subpopulation cell numbers. Taken together this demonstrates macrophage heterogeneity in *Drosophila*, identifies mechanisms involved in subpopulation specification and function and facilitates the use of *Drosophila* to study macrophage heterogeneity in vivo.

**\*For correspondence:**
i.r.evans@sheffield.ac.uk

**Competing interests:** The authors declare that no competing interests exist.

## Introduction

Macrophages are key innate immune cells responsible for clearing infections, debris, and apoptotic cells, the promotion of wound healing and are necessary for normal development (*Wynn et al., 2013*). However, their aberrant behaviour can also cause or exacerbate numerous human disease states, including cancer, atherosclerosis, and neurodegeneration (*Wynn et al., 2013*). Macrophages are a highly heterogeneous population of cells, which enables them to carry out their wide variety of roles, and this heterogeneity arises from diverse processes. These processes include the dissemination and maintenance of tissue resident populations (*Gordon and Plüddemann, 2017*) and the ability to adopt a spectrum of different activation states (termed macrophage polarisation), which can range from pro-inflammatory (historically termed as M1-like) to anti-inflammatory, pro-healing (M2-like) macrophage activation states (*Martinez and Gordon, 2014*; *Murray, 2017*).

Macrophage heterogeneity is not limited to mammals, appearing conserved across vertebrate lineages – both in terms of polarisation and the presence of tissue resident populations. For example, evidence suggests the existence of pro-inflammatory macrophage populations in zebrafish (*Nguyen-Chi et al., 2015*), with polarisation also a well-defined phenomenon in other fish species (*Wiegertjes et al., 2016*). Zebrafish are also known to contain tissue resident macrophages such as myeloid-derived microglia (*Ferrero et al., 2018*; *Xu et al., 2016*). Vertebrate macrophages interact with and can become polarised in response to signals produced by Th1 and Th2 cells, leading to acquisition of M1-like and M2-like activation states, respectively (*Murray, 2017*), while B- and T-cell-based adaptive immunity is thought to have evolved in teleost fish (*Buchmann, 2014*). Therefore, the absence of an adaptive immune system may restrict the diversity of macrophage populations in

more simple organisms that possess only an innate immune system. However, the fact that macrophage markers can be highly divergent, even when comparing mammals as closely related as mice and humans (*Murray and Wynn, 2011*), has hampered investigation of whether this is indeed the case, indicating a need for alternative markers and approaches.

Macrophage heterogeneity has been extensively studied in mammalian systems and, although this has provided a good understanding of how macrophages determine their polarisation state, this has also identified considerable complexity with many activation states possible (*Murray et al., 2014*). Additional complexity arises with both M1-like and M2-like macrophages found at the same sites of pathology, for example within atherosclerotic plaques (*Colin et al., 2014*). Furthermore, the cytokine profiles that can be induced in vitro depend on the exact activation methods used experimentally and these do not necessarily reflect polarisation states in vivo (*Vogel et al., 2014*), while other macrophage subpopulations may be missed by in vitro approaches. Given these intricacies, it is clear that we need to better understand the fundamental components and pathways responsible for the specification of different macrophage subtypes, particularly in vivo. Recently, the 'macrophage-first' hypothesis has been proposed, re-emphasising the idea that acute signals can polarise macrophages ahead of the involvement of T cells (*Wiegertjes et al., 2016*). Consequently, organisms without a fully developed adaptive immune system represent intriguing models in which to examine this idea and better understand macrophage heterogeneity in vivo.

*Drosophila melanogaster* has been extensively utilised to study innate immunity (*Buchon et al., 2014*), but lacks an adaptive immune system. Fruit fly blood is specified in two waves – an embryonic wave in the head mesoderm and in the larval lymph gland, with those cells released at the end of larval development (*Gold and Brückner, 2015*). Blood cell proliferation has also been shown to occur in haematopoietic pockets attached to the larval body wall (*Leitão and Sucena, 2015*; *Makhijani et al., 2011*). These waves of haematopoiesis generate three types of blood cell (also referred to as hemocytes): plasmatocytes, crystal cells, and lamellocytes. Of these, plasmatocytes are functionally equivalent to vertebrate macrophages (*Evans et al., 2003*; *Wood and Jacinto, 2007*), with the capacity to phagocytose apoptotic cells and pathogens, secrete extracellular matrix, disperse during development and migrate to sites of injury (*Ratheesh et al., 2015*). Although *Drosophila* blood lineages are considerably less complex than their vertebrate equivalents, they are specified via transcription factors related to those used during vertebrate myelopoiesis, including GATA and Runx-related proteins (*Evans et al., 2003*). Furthermore, plasmatocytes utilise evolutionarily conserved genes in common with vertebrate innate immune cells to migrate (e.g. SCAR/WAVE, integrins, and Rho GTPases [*Comber et al., 2013*; *Evans et al., 2013*; *Paladi and Tepass, 2004*; *Siekhaus et al., 2010*; *Stramer et al., 2005*]) and phagocytose (e.g. the CED-1 family member Draper [*Manaka et al., 2004*] and CD36-related receptor Croquemort [*Franc et al., 1996*]). Given these striking levels of functional and molecular conservation, *Drosophila* has been extensively used for research into macrophage behaviour in vivo with its genetic tractability and in vivo imaging capabilities facilitating elucidation of different macrophage behaviours conserved through evolution (*Ratheesh et al., 2015*; *Wood and Jacinto, 2007*). However, despite these evolutionarily-conserved commonalities, the plasmatocyte lineage has, until very recently, been considered a homogeneous cell population. Hints that *Drosophila* plasmatocytes may exhibit heterogeneity exist in the literature with variation in marker expression observed in larval hemocytes (*Anderl et al., 2016*; *Kurucz et al., 2007a*; *Shin et al., 2020*) and non-uniform expression of TGF-β homologues upon injury or infection in adults (*Clark et al., 2011*). Recent single-cell RNA-sequencing (scRNAseq) experiments performed on larval hemocytes have also suggested the presence of multiple clusters of cells, which were interpreted as representing either different stages of differentiation or functional groupings (*Cattenoz et al., 2020*; *Tattikota et al., 2019*). However, the in vivo identification of subtypes and insights into the roles and specification mechanisms of potential macrophage subtypes in *Drosophila* has not yet been described.

Here, we describe the first identification and characterisation of molecularly and functionally distinct plasmatocyte subpopulations within *Drosophila melanogaster*. Drawing on a collection of reporter lines (https://enhancers.starklab.org/; *Kvon et al., 2014*), we have identified regulatory elements that define novel plasmatocyte subpopulations in vivo. We show that these molecularly distinct subpopulations exhibit functional differences compared to the overall plasmatocyte population and that the proportion of cells within these subpopulations can be modulated by external stimuli such as increased levels of apoptosis. Furthermore, we show that misexpression of a

gene associated with a subpopulation-specific enhancer element is able to modulate plasmatocyte behaviour in vivo, thereby identifying novel effector genes of plasmatocyte subpopulation function. Together our findings reveal that macrophage heterogeneity is a fundamental and evolutionarily conserved characteristic of innate immunity that pre-dates the development of the adaptive immune system. This significantly extends the utility of an already powerful genetic model system and provides further avenues to understand regulation of innate immunity and macrophage heterogeneity.

## Results

### *Drosophila* embryonic plasmatocytes do not behave as a uniform population of cells

The macrophage lineage of hemocytes (plasmatocytes) has historically been considered a homogeneous population of cells. However, careful analysis of plasmatocyte behaviour in vivo suggested to us that this lineage might not be functionally uniform. For instance, imaging the inflammatory responses of plasmatocytes to epithelial wounds, we find that some cells close to injury sites rapidly respond by migrating to the wound, while other neighbouring cells fail to respond (*Figure 1a*; *Video 1*). We also find that plasmatocytes exhibit variation in their expression of well-characterised plasmatocyte markers such as *crq-GAL4* (*Figure 1b–b'*; *Franc et al., 1996*; *Stramer et al., 2005*) and display a broad diversity in their migration speeds within the embryo (random migration at stage 15; *Figure 1c–d*). These professional phagocytes also display differences in their capacities to phagocytose apoptotic cells with some cells engulfing many apoptotic particles, whereas others engulf very few, if any (*Figure 1e*). Furthermore, phagocytosis of microorganisms by larval hemocytes also varies significantly from cell-to-cell in vitro (*Figure 1f*). These differences within the plasmatocyte lineage led us to hypothesise that this cell population is more heterogeneous than previously appreciated.

### Discrete subpopulations of plasmatocytes are present in the developing *Drosophila* embryo

Given the diversity in plasmatocyte behaviour observed (*Figure 1*), we hypothesised that macrophage heterogeneity represents an evolutionarily conserved feature of innate immunity, which therefore originally evolved in the absence of an adaptive immune system. To address this and look for molecular differences between plasmatocytes, we examined transgenic enhancer reporter lines (*VT-GAL4* lines) produced as part of a large-scale tilling array screen (*Kvon et al., 2014*) that had been annotated as labelling hemocytes (http://enhancers.starklab.org/). Based on examination of the published *VT-GAL4* expression patterns, we identified *VT-GAL4* lines that appeared to label reduced numbers of plasmatocytes in the embryo, reasoning that plasmatocyte subpopulations could be molecularly identified on the basis of differences in reporter expression. While a number of the enhancers appeared to label all plasmatocytes (e.g. *VT41692-GAL4*), we identified several that labelled discrete numbers of plasmatocytes (*Figure 2a*). We next confirmed that the cells labelled by these *VT-GAL4* lines were plasmatocytes by using these constructs to drive expression of *UAS-tdTomato* in the background of a *GAL4*-independent, pan-hemocyte marker (*srpHemo-GMA – serpent* enhancer region driving expression of a GFP-tagged actin-binding domain of Moesin; *Figure 2b–d*). As initially predicted based on their morphology and position during embryogenesis, each of the *VT-GAL4* lines marking potential subpopulations did indeed express in the hemocyte lineage (*Figure 2e*). These subpopulation cells were identified as plasmatocytes based upon their morphology, the absence of lamellocytes in embryos and the non-migratory nature of crystal cells (*Wood and Jacinto, 2007*) and could be observed to follow both the dorsal and ventral migration routes (*Ratheesh et al., 2015*) used by plasmatocytes during their developmental dispersal (*Figure 2e*). In order to quantify the proportion of cells labelled by each *VT-GAL4* line, we counted the number of cells labelled on the ventral midline of the developing stage 15 embryo, using *VT-GAL4* lines to drive expression from *UAS-GFP*. This verified reproducible and consistent labelling of discrete subsets of plasmatocytes (*Figure 2f–h*), suggesting that these cells represent stable subpopulations within this macrophage lineage.

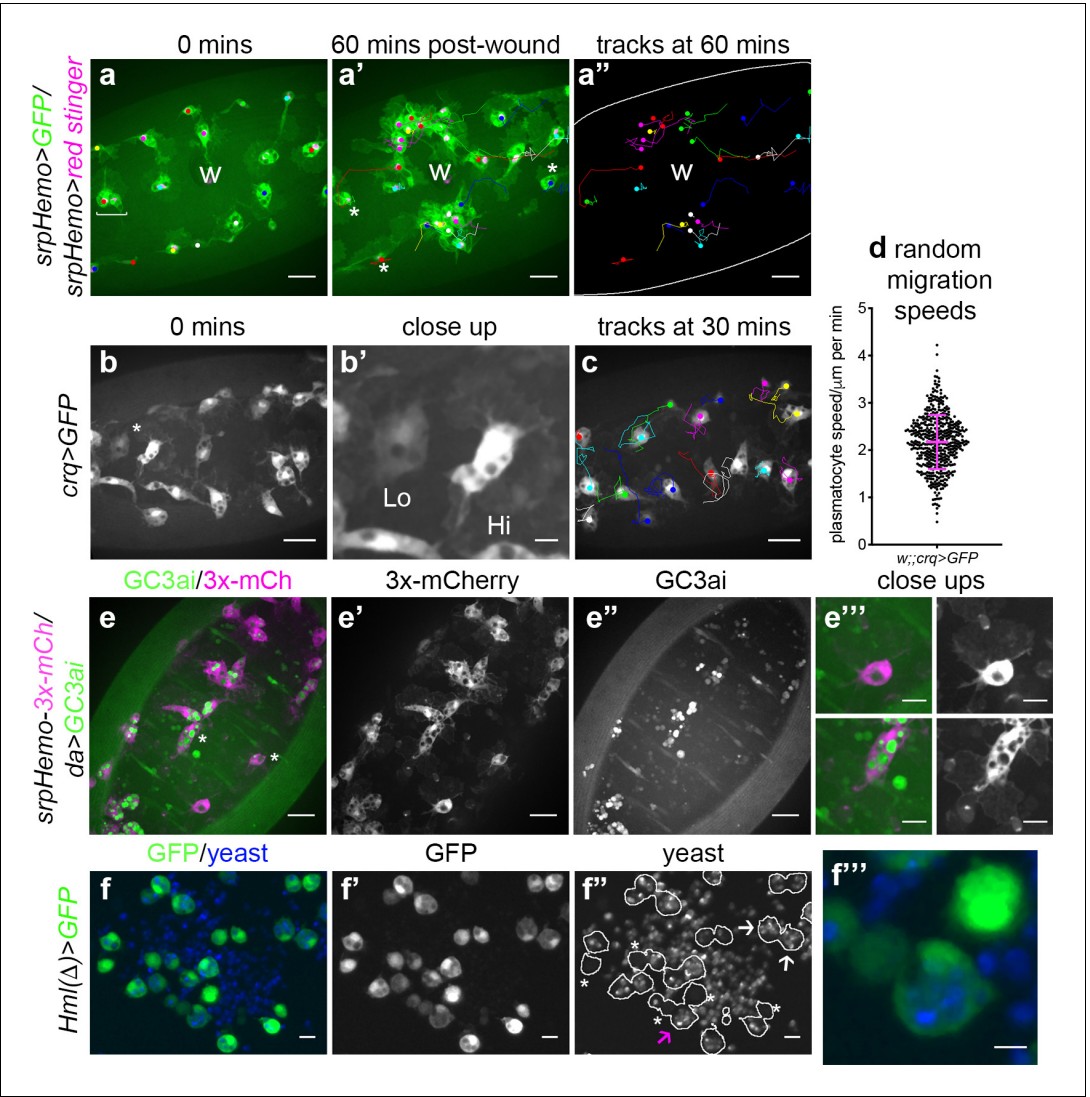

**Figure 1.** Heterogeneity of *Drosophila* embryonic plasmatocyte responses. (a) GFP (green) and nuclear red stinger (magenta) labelled plasmatocytes on the ventral side of a stage 15 embryo at 0 min (a) and 60 min post-wounding (a'); plasmatocyte tracks at each timepoint are overlaid (a–a') or shown in full (a''). Examples of plasmatocytes failing to respond to the wound (w) indicated via asterisks; square bracket (a) indicates neighbouring plasmatocytes, one of which responds to wounding, while the other fails to respond (see *Video 1*). (b) Imaging of plasmatocytes labelled using *crq-GAL4* to drive expression of GFP reveals a wide range in levels of *crq* promoter activity within plasmatocytes at stage 15; (b') Close-up of cells marked by an asterisk in (b). (c) Overlay of plasmatocyte tracks of cells shown in (b) showing significant variation in their random migration speeds. (d) Scatterplot of plasmatocyte random migration speeds (taken from 23 embryos); line and error bars show mean and standard deviation, respectively. (e) Imaging the ventral midline at stage 15 shows a wide range in the amount of apoptotic cell clearance (green in merge; labelled via the caspase-sensitive reporter GC3ai) undertaken by plasmatocytes (magenta in merge, labelled via *srpHemo-3x-mCherry* reporter); (e'–e'') mCherry and GC3ai channels; (e''') close-ups of cells devoid/full of engulfed GC3ai particles (indicated by asterisks in (e)). (f) Larval hemocytes (green in merge, labelled via *Hml(Δ)-GAL4*-driven expression of GFP) exhibit a range in their capacities to engulf calcofluor-labelled yeast (blue in merge) in vitro; (f'–f'') GFP and yeast channels; white lines indicate cell edges in (f''); asterisks in (f'') indicate cells that have failed to phagocytose yeast; white arrows in (f'') indicate cells that have phagocytosed multiple yeast particles; magenta arrow in (f'') indicates close-up of region indicated in (f'''). Scale bars represent 20 μm (a–a'', b, c, e–e''), 10 μm (e''', f–f''), or 5 μm (b', f'''). See *Supplementary file 1* for full list of genotypes.

The online version of this article includes the following source data for figure 1:

**Source data 1.** Numerical data used to plot panel (d) of *Figure 1*.

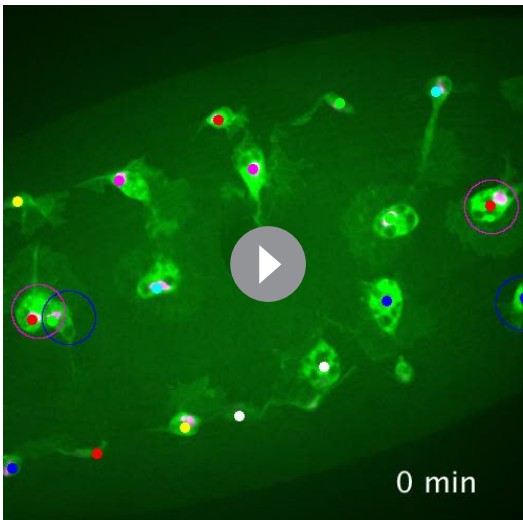

**Video 1.** Plasmatocytes in similar positions within the embryo do not respond equally to inflammatory stimuli. GFP (green) and red stinger (magenta) labelled plasmatocytes responding to an epithelial wound at stage 15. Tracks of cell movements are shown via dots and lines. Magenta circles show cells responding to the wound; blue circles indicate cells that are the same distance from the wound but fail to respond to the wound. Movie corresponds to stills shown in *Figure 1a* and lasts for 60 min post-wounding. Scale bar represents 20 μm. See *Supplementary file 1* genotype in full.

https://elifesciences.org/articles/58686#video1

To characterise these subpopulations further, their overlap with the known plasmatocyte markers Eater, Croquemort, and Simu was investigated using novel GAL4-independent *VT-RFP* reporters, which we generated using the same enhancer sequences and insertion sites originally characterised by the Stark lab (*Kvon et al., 2014*). While we were unable to detect embryonic expression of *eater* at this stage of development (*Figure 2—figure supplement 1*), reporters for *crq* and *simu* (*crq-GAL4,UAS-GFP* and *simu-cytGFP*), which also encode phagocytic receptors (*Franc et al., 1996*; *Kurant et al., 2008*), clearly labelled embryonic plasmatocytes (*Figure 2—figure supplement 2*). *simu-cytGFP*-labelled plasmatocytes with little cell-to-cell variation (*Figure 2—figure supplement 2a*), whereas *crq-GAL4,UAS-GFP* displayed considerable heterogeneity (*Figure 1b–b'*; *Figure 2—figure supplement 2b–c*). However, there was little correlation of *simu* or *crq* marker expression with subpopulation cells, since all cells expressed similar levels of *simu*, while both *crq* and *VT-RFP* expression appeared to vary independently of each other (*Figure 2—figure supplement 2a–b*). Taken together, we were able to detect discrete subpopulations of plasmatocytes in the embryo, but these subpopulations showed no clear segregation with existing plasmatocyte markers.

## Subpopulations of *Drosophila* plasmatocytes vary across development: subpopulation dynamics in larvae and white pre-pupae

Having identified subpopulations of plasmatocytes in the embryo, we then examined other stages of development to see whether their presence was maintained or modulated over time. In order to exclude potential expression in non-hemocyte cells (e.g. the non-plasmatocyte cells apparent in *Figure 2e*), we labelled subpopulation cells specifically using a split GAL4 approach (*Pfeiffer et al., 2010*), employing the *serpent* enhancer (a well-characterised hemocyte marker; *Lebestky et al., 2000*; *Rehorn et al., 1996*) and VT enhancers to express the transcriptional activation domain (AD) and DNA binding domains (DBD) of GAL4 independently. Only when co-expressed in the same cell do the AD and DBD heterodimerise and allow expression of UAS transgenes (*Figure 3—figure supplement 1a*). Characterising the split GAL4 lines in the embryo via expression of the EGFP-derivative Stinger (*Barolo et al., 2000*) confirmed that this split GAL4 approach labels discrete subpopulations of plasmatocytes within the embryo, although with a higher proportion of cells labelled compared to the original *VT-GAL4* lines (*Figure 3—figure supplement 1b–d*) – a difference likely due to a combination of amplification via the split GAL4 system and enhanced detectability of Stinger, which accumulates in the nucleus due to its nuclear localisation signal. Similar trends in the proportions of plasmatocytes labelled in a variety of locations across the embryo were observed for each subpopulation; for example, *VT32897*-labelled cells were the least frequently observed in the lateral head region and on the dorsal and ventral sides of the embryo (*Figure 3—figure supplement 1b–d*). As per *Figure 2e*, this suggests there are no clear biases between the dispersal routes undertaken by subpopulation cells.

While *serpent* expression decreases in hemocytes during larval stages, we found that *srpHemo-AD;srpHemo-DBD* in concert with *UAS-stinger* was sufficient to label large numbers of cells in both L1 and L2 larvae (*Figure 3a–b*), consistent with previous publications (*Gyoergy et al., 2018*).

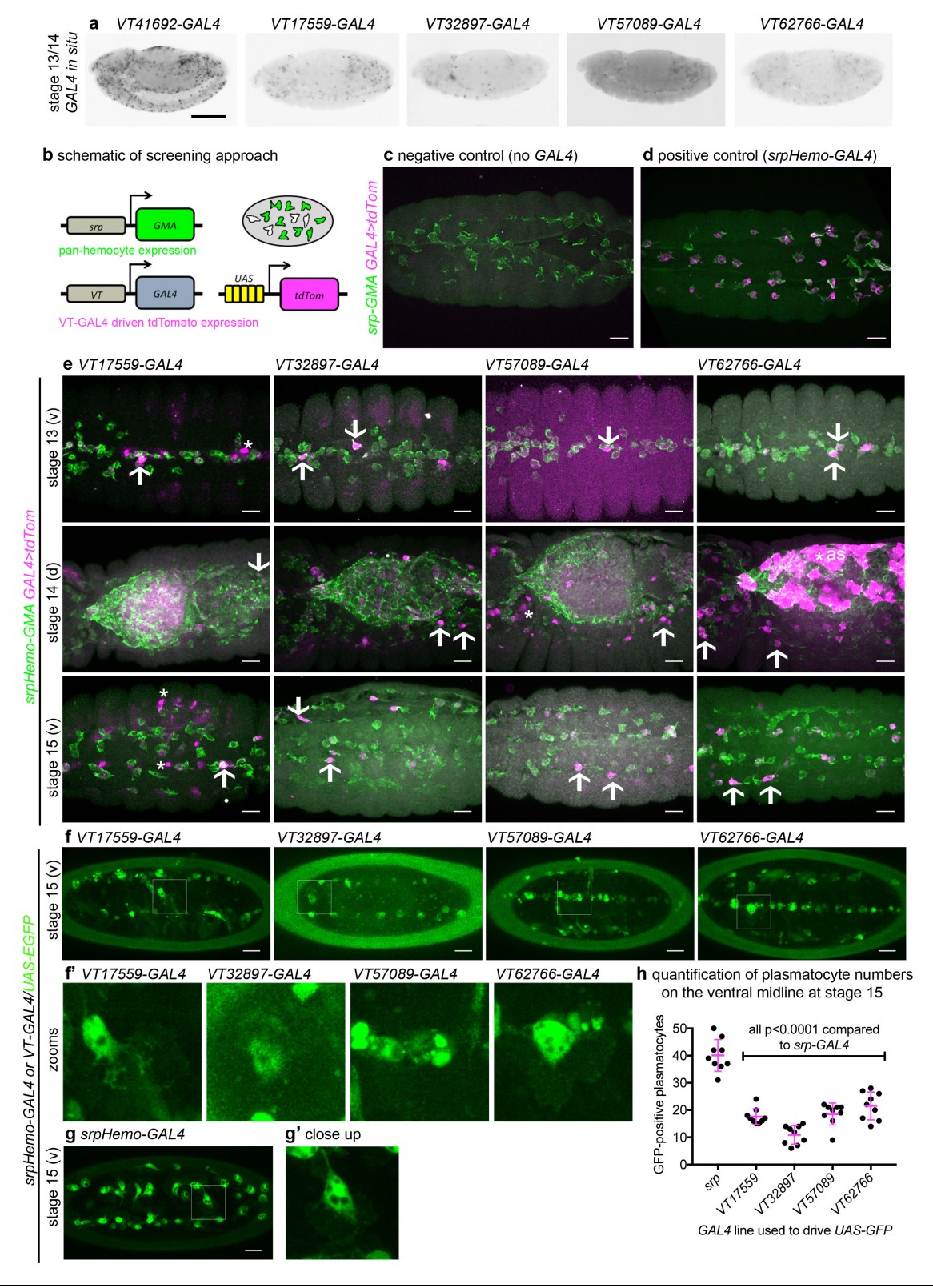

**Figure 2.** Identification of enhancers labelling discrete plasmatocyte subpopulations in *Drosophila*. (a) Lateral views of stage 13/14 embryos with in situ hybridisation performed for GAL4 for indicated *VT-GAL4* lines (anterior is left). Taken with permission from http://enhancers.starklab.org/ (n.b. these images are not covered by the CC-BY 4.0 licence and further reproduction of this panel would need permission from the copyright holder); *VT41692-GAL4* represents an example in which the majority of plasmatocytes are labelled. (b) Schematic diagram showing screening approach to identify

*Figure 2 continued on next page*

*Figure 2 continued*

subpopulations of plasmatocytes: *VT-GAL4*-positive plasmatocytes will express both GMA (green) and tdTomato (magenta) – white cells in the schematic. (c–d) Images showing the ventral midline at stage 14 of negative control (no driver; *w;UAS-tdTom/+;srpHemo-GMA*) and positive control (*w; srpHemo-GAL4/UAS-tdTom;srpHemo-GMA*) embryos. (e) Images showing embryos containing *VT-GAL4*-labelled cells (via *UAS-tdTomato*, shown in magenta) at stage 13 (first row, ventral views), stage 14 (second row, dorsal views), and stage 15 (third row, ventral views). The entire hemocyte population is labelled via *srpHemo-GMA* (green); arrows indicate examples of *VT-GAL4*-positive plasmatocytes; asterisks indicate *VT-GAL4*-positive cells that are not labelled by *srpHemo-GMA*. N.b. *VT62766-GAL4* image contrast enhanced to different parameters compared to other images owing to the very bright labelling of amnioserosal cells (cells on dorsal side of embryo destined to be removed during dorsal closure; labelled with an asterisk) in the stage 14 image. (f) Labelling of smaller numbers of plasmatocytes on the ventral midline at stage 15 using *VT-GAL4* lines indicated and *UAS-GFP* (green); boxed regions show close-ups of *VT-GAL4*-positive plasmatocytes (f'). (g) Ventral view of positive control embryo (*w;srpHemo-GAL4,UAS-GFP*) and example plasmatocyte (g') at stage 15. (h) Scatterplot showing numbers plasmatocytes labelled using *VT-GAL4* lines to drive expression from *UAS-GFP* on the ventral midline at stage 15; lines and error bars represent mean and standard deviation, respectively. p-Values calculated via one-way ANOVA with a Dunnett's multiple comparison post-test (all compared to *srpHemo-GAL4* control); n = 9 embryos per genotype. Scale bars represent 150 µm (a) or 10 µm (c–g). See *Supplementary file 1* for full list of genotypes; overlap of VT enhancer expression with known plasmatocyte markers can be found in *Figure 2—figure supplements 1* and *2*.

The online version of this article includes the following source data and figure supplement(s) for figure 2:

**Source data 1.** Numerical data used to plot panel (h) of *Figure 2*.
**Figure supplement 1.** Subpopulation cells do not express *eater* in the embryo.
**Figure supplement 2.** *crq* and *simu* do not specifically mark subpopulation cells in the developing embryo.
**Figure supplement 2—source data 1.** Numerical data used to plot panel (c) of *Figure 2—figure supplement 2*.

Following hatching of embryos, we cannot use cell morphology to discriminate between plasmatocytes and other hemocyte lineages (crystal cells and lamellocytes) and therefore refer to subpopulation cells as hemocytes for post-embryonic stages of development. Quantification of the numbers of subpopulation cells that could be detected using the split GAL4 system to drive expression from *UAS-stinger* showed that roughly 50% of *serpent*-positive hemocytes were labelled in L1 larvae for each VT subpopulation (*Figure 3a–c*). We cannot exclude the possibility that some of the cells labelled in L1 larvae are fat body cells as *serpent* is known to be expressed in the fat body (*Rehorn et al., 1996*). Therefore, it is possible that greater than 50% of hemocytes are labelled at this stage; nonetheless, a significant proportion of subpopulation cells are not labelled via these split GAL4 reporters in L1 larvae (*Figure 3a–c*). At this stage of development, most hemocytes are found in sessile patches attached to the body wall (*Lanot et al., 2001*; *Makhijani et al., 2011*) – this is also the case for the majority of subpopulation cells, since live imaging shows little movement relative to other cells during larval crawling, although some circulating cells could be observed (*Video 2*).

In contrast to the significant numbers of cells present in L1 and L2 larvae, imaging of L3 larvae containing split *srpHemo-AD* and *VT-DBD* reporters (abbreviated to *VTn*) revealed that fewer subpopulation cells could be detected at this stage (*Figure 4a–f*). This decrease in subpopulation cells does not seem to be linked to lower levels of *serpent* expression because blood cells are robustly labelled in positive control L3 larvae (*srpHemo-AD* in combination with *srpHemo-DBD*; *Figure 4b*), suggesting that serpent expression is not limiting our ability to detect subpopulation cells. Moreover, a reduction in subpopulation cells can also be seen when using the original *VT-GAL4* lines to drive stinger expression at this stage (i.e. independent of a reliance on *serpent* expression for labelling; *Figure 4—figure supplement 1a*).

Using this approach, *VT32897* and *VT17559* labelled the most cells (*Figure 4c–d*), with only the occasional cell present in *VT57089* larvae (*Figure 4e*) and cells largely absent from *VT62766* larvae (*Figure 4f*). Labelled cells were also present in the head region, along the dorsal vessel (the fly heart) and between the salivary glands (which themselves exhibit non-specific labelling) in *VT32897* larvae. The *VT32897* head region cells are likely to represent sessile hemocytes, whereas cells at the remaining two sites probably correspond to *serpent*-positive nephrocytes and garland cells (*Brodu et al., 1999*; *Das et al., 2008*), respectively (*Figure 4d*). *VT57089* shows additional staining in the head region (potentially the Bolwig organ; *Figure 4e*) and, as per the dorsal vessel-associated cells in *VT32897* (*Figure 4d*), hemocytes can also be found in these regions when the total hemocyte population is labelled using *srpHemo-AD* and *srpHemo-DBD* in positive controls (*Figure 4b*; *Video 3*). Furthermore, these larval distributions closely resemble patterns observed using *VT-GAL4* reporters, albeit with a loss of non *srp*-dependent labelling due to our split GAL4 approach (data not shown).

Live imaging of L3 larvae confirmed that hemocytes were predominantly attached to the body wall, but that small numbers of cells could be detected in circulation (*Video 4*).

To quantify the proportion of hemocytes that were labelled in L3 larvae and address any bias in localisation, L3 larvae were bled and then the carcasses scraped (as described in *Petraki et al., 2015*) to compare numbers of cells in circulation with those more tightly adhered to tissue, respectively. This approach confirmed the low numbers of cells observed in L3 larval images and revealed no bias in subpopulation localisation, with similar proportions present in circulation and adhered to tissue for each subpopulation (*Figure 4g*). Quantification of subpopulation localisation along the L3 body axis suggested that *VT57089* and *VT62766* cells exhibit a bias toward the posterior of the larvae compared to the total population (*Figure 4h–h'*).

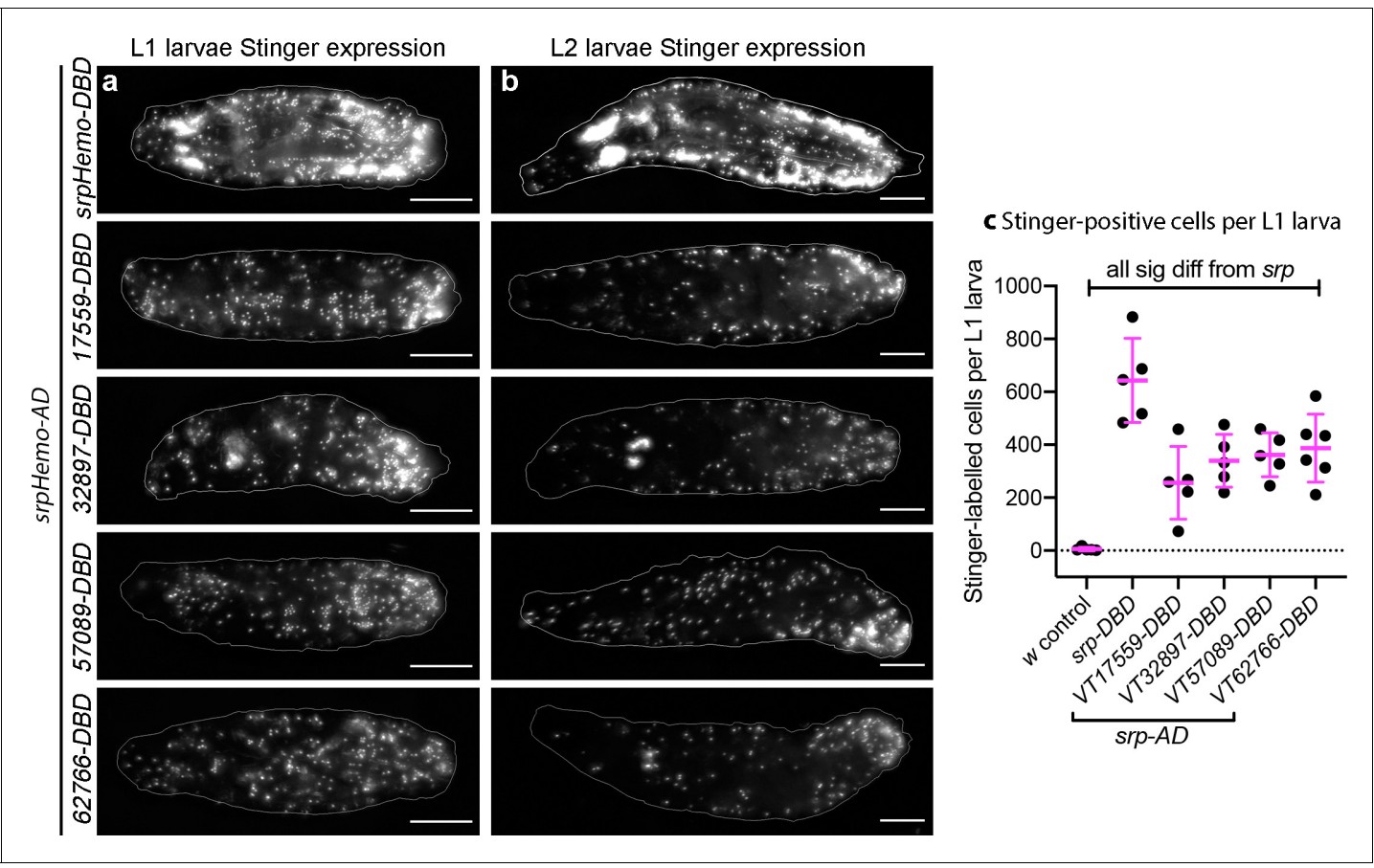

**Figure 3.** Plasmatocyte subpopulations are present in large numbers in L1 and L2 larvae. (a–b) Images of L1 (a) and L2 larvae (b) with cells labelled using the split GAL4 system (*srpHemo-AD* in combination with *srpHemo-DBD* or the *VT-DBD* transgene indicated) to drive expression from *UAS-stinger*. Scale bars represent 150 μm; white lines show edge of the larva; images contrast enhanced to 0.3% saturation. (c) Scatterplot showing numbers of Stinger-positive cells labelled via the split GAL4 system per larva; numbers of cells were quantified from flattened L1 larvae. $w^{1118}$;*UAS-stinger/+* larvae were used as negative controls; all conditions are significantly different compared to the positive control ($w^{1118}$;*srpHemo-AD/UAS-stinger; srpHemo-DBD/+*) via a one-way ANOVA with a Dunnett's multiple comparison post-test: *srp* vs *w*, p<0.0001; *srp* vs *VT17559* p<0.0001; *srp* vs *VT32897*, p=0.0013; *srp* vs *VT57089*, p=0.0029; *srp* vs *VT62766*, p=0.0047; n = 5 for *w* control, *srp*, *VT17559*, *VT32897*, and *VT57089* and n = 6 for *VT62766*. See *Supplementary file 1* for full list of genotypes; a schematic and validation of this split GAL4 approach in the embryo can be found in *Figure 3—figure supplement 1*.

The online version of this article includes the following source data and figure supplement(s) for figure 3:

**Source data 1.** Numerical data used to plot panel (c) of *Figure 3*.

**Figure supplement 1.** Using a split GAL4 approach to label plasmatocyte subpopulations.

**Figure supplement 1—source data 1.** Numerical data used to plot panel (e) of *Figure 3—figure supplement 1*.

**Figure supplement 1—source data 2.** Numerical data used to plot panel (f) of *Figure 3—figure supplement 1*.

**Figure supplement 1—source data 3.** Numerical data used to plot panel (g) of *Figure 3—figure supplement 1*.

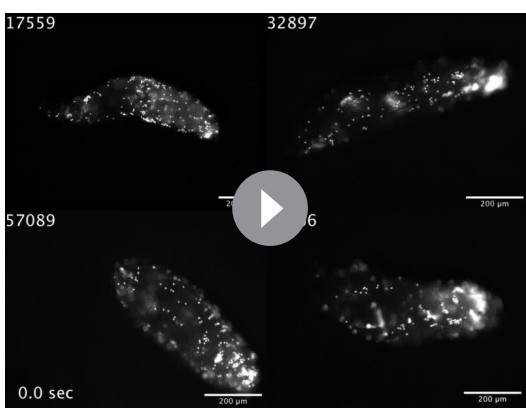

**Video 2.** Localisation and dynamics of subpopulation cells within L1 larvae. Movies showing localisation and movement of cells labelled using split GAL4 lines (*srpHemo-AD* in combination with *VT17559-DBD*, *VT32897-DBD*, *VT57089-DBD*, or *VT62766-DBD*) to drive expression from *UAS-stinger* in L1 larvae. Images taken from timelapse series of single focal planes to enable rapid imaging. The majority of cells detected appear attached to the body wall, since they do not shift their relative positions during larval movements, although some cells can be seen in circulation. Scale bars represent 200 µm. See *Supplementary file 1* for full list of genotypes.

https://elifesciences.org/articles/58686#video2

The striking decrease in proportion of subpopulation cells that can be detected in L3 larvae compared to earlier stages suggests reprogramming of cells leading to a loss of enhancer activity. To test this hypothesis, split GAL4 lines were used in lineage tracing experiments via G-TRACE (*Evans et al., 2009*). In this approach, current expression of reporters is marked via Red Stinger and also leads to the permanent expression of Stinger via the activity of a co-expressed FLP recombinase. As such, the expression of both fluorophores indicates current split GAL4 activity, while the expression of Stinger alone indicates historical activity in cells where expression has since ceased (*Figure 4—figure supplement 1b*). Analysing the total complement of current and historic subpopulation cells in L3 larvae via G-TRACE revealed that a significant proportion of subpopulation cells were positive for historical expression only (Stinger expression alone; *Figure 4—figure supplement 1c*). This confirms that subpopulation cells from earlier timepoints survive in vivo but change their transcriptional profile – a finding consistent with reprogramming events rather than a loss of cells themselves.

Imaging of white pre-pupae (WPP), the stage that marks the beginning of pupal development and metamorphosis, showed very similar patterns across the split GAL4 VT enhancer lines (*Figure 5a–f*), with a further reduction in the numbers of cells labelled. It was possible to observe the occasional cell moving in circulation within WPP, strongly suggesting these cells are hemocytes (*Video 5* and *Video 6*). Live imaging of *VT32897* WPP also confirmed association of cells with the pumping dorsal vessel (*Figure 5d*; *Video 7*). Significantly, this data indicates that the presence of subpopulations within embryos is not simply a consequence of slow accumulation of fluorescent proteins by weak drivers, since these enhancer-based reporters do not label an ever-increasing number of cells as development proceeds. Overall, the numbers of hemocytes within subpopulations that can be detected decreases over larval and early pupal stages, suggesting that plasmatocyte subpopulations are developmentally regulated and exhibit plasticity. This reprogramming could reflect specific and changing requirements for specialised plasmatocyte subpopulations across the life cycle, for example, an association with processes required for organogenesis (*Charroux and Royet, 2009*; *Defaye et al., 2009*; *Regan et al., 2013*). The differential localisation of some subpopulation cells also indicates the potential that molecularly and functionally different macrophage populations are present at specific tissues in the fly.

## Subpopulation cells return in large numbers during pupal development

Since subpopulation cells appear associated with stages of development when organogenesis and tissue remodelling occur, we hypothesised that some hemocytes may be reprogrammed via changes in expression leading to reactivation of the enhancers that mark these subpopulations. This would enable subpopulations to return during metamorphosis. Imaging pupae at various times after puparium formation (APF) revealed that subpopulation cells re-emerged in large numbers during this stage, but with distinct dynamics between subpopulations labelled with different enhancers (*Figure 6a–f*). For instance, *VT17559* cells are already present in substantial numbers by 18 hr APF (*Figure 6c*), whereas *VT32897* reporter expression reappears between 24 and 48 hr APF (*Figure 6d*). *VT57089* and *VT62766* cells increased in numbers more gradually over the course of pupal development (*Figure 6e–f*). Different subpopulations appear present in subtly distinct

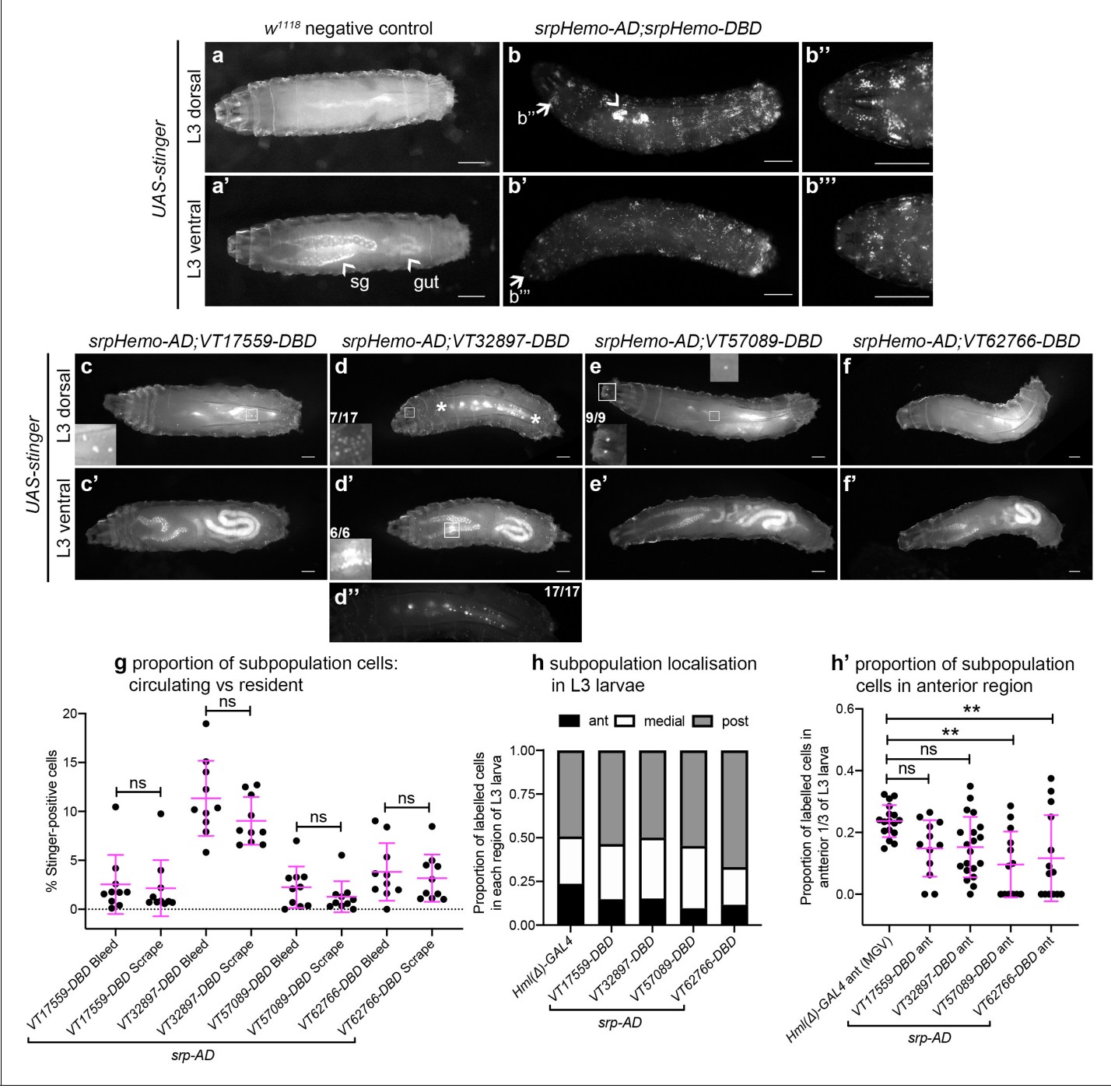

**Figure 4.** Plasmatocyte subpopulations are greatly reduced in L3 larvae but exhibit distinctive localisations. (a–f) Dorsal and ventral views of negative control L3 larvae (a, no GAL4), positive control L3 larvae with hemocytes labelled via *serpent* (b, *UAS-stinger* driven by *srpHemo-AD;srpHemo-DBD*) and L3 larvae containing cells labelled through expression of *UAS-stinger* via *srpHemo-AD* and the *VT-DBD* transgenes indicated (c–f). Arrowheads indicate non-specific expression of Stinger in salivary glands and gut (a' – also visible in dorsal images (c'–f') but not labelled) and possible proventricular region hemocytes/garland cells (b); arrows (b, b') indicate regions shown in close-ups of potential hemocyte population in the head region (b'') and in the Bolwig organ (b'''); boxes indicate individual hemocytes (c, e) and labelling in the head region (d), proventriculus/of Garland cells (d'), and Bolwig organ (e) shown at enhanced magnification in inset panels; asterisks in (d) denote region shown as a close-up and at a reduced brightness in (d'') in order to reveal detail of cells along the dorsal vessel; fractions indicate the number of larvae exhibiting a particular localisation out of the total imaged. (g) Scatterplot showing the proportion of subpopulation cells labelled via the split GAL4 system in circulation (initial bleed) compared to the proportions in resident/adhered populations (scraping of the carcass) in the indicated genotypes. Proportions obtained via each method compared via Student's t-test (n = 10 larvae per genotype; p=0.77 (*VT17559*), p=0.13 (*VT32897*), p=0.27 (*VT57089*), p=0.60 (*VT62766*)). (h) Bar chart showing the relative proportions of

*Figure 4 continued on next page*

*Figure 4 continued*

labelled cells found within the anterior, medial or posterior 1/3 of L3 larvae using *Hml(Δ)-GAL4* to drive EGFP or the split GAL4 system to express Stinger in all larval hemocytes or subpopulations, respectively (n = 17, 12, 20, 13, 14 larvae). (h') Scatterplot of the proportions of cells found within the anterior region of L3 larvae for controls and split GAL4 lines. Kruskall-Wallis test with Dunn's multiple comparisons test was used to compare subpopulation values with *Hml(Δ)-GAL4* control; (p=0.11 (*VT17559*), p=0.061 (*VT32897*), p=0.0018 (*VT57089*), p=0.0063 (*VT62766*)). Scale bars represent 500 µm (a–f); larval images contrast enhanced to 0.3% saturation (a–f); lines and error bars represent mean and standard deviation, respectively (g, h'); bars represent mean (h); ns and ** denote not significant and p<0.01, respectively. See *Supplementary file 1* for full list of genotypes; see *Figure 4—figure supplement 1* for quantification of numbers of subpopulation cells labelled using the original *VT-GAL4* lines and lineage tracing of subpopulation cells via G-TRACE.

The online version of this article includes the following source data and figure supplement(s) for figure 4:

**Source data 1.** Numerical data used to plot panel (g) of *Figure 4*.
**Source data 2.** Numerical data used to plot panels (h) and (h') of *Figure 4*.
**Figure supplement 1.** Lineage tracing shows reprogramming of subpopulation cells in L3 larvae.
**Figure supplement 1—source data 1.** Numerical data used to plot panel (a) of *Figure 4—figure supplement 1*.
**Figure supplement 1—source data 2.** Numerical data used to plot panel (c) of *Figure 4—figure supplement 1*.

locations in pupae (*Figure 6*). Further work will be required to understand if subpopulation specification occurs in situ or cells are specified and then migrate to these regions.

## Subpopulations display distinct dynamics and localisation in adults

Immediately after adults hatch, large numbers of split GAL4-labelled cells can be observed across all lines and are present in selected regions that overlap with the overall adult hemocyte population (*Figure 7a–e*). The overall hemocyte population remains detectable as adults age (0–6 weeks; *Figure 7a*); however, not all subpopulations exhibit an identical localisation or dynamics during this time (*Figure 7b–e*). *VT57089* and *VT62766* cells largely disappear by 1 week (*Figure 7d–e*) and the majority of *VT17559*-labelled cells are absent by 2 weeks (*Figure 7b*). By contrast, *VT32897* cells can be detected for at least 6 weeks of adult life and are particularly prominent in the thorax at 4 weeks (*Figure 7c*). Other differences in localisation are also apparent with cells particularly obvious in the legs for the *VT17559* line (*Figure 7b*, day 1–2 weeks), whereas *VT57089* and *VT62766*-labelled cells are more closely associated with the thorax and dorsal abdomen (*Figure 7d–e*, day 1). Labelled cells are also present in the proboscis for several lines (*Figure 7c–e*).

To quantify the proportion of blood cells labelled in adults, 1-day-old flies were dissected. Despite the large numbers of cells labelled via the split GAL4 system in adults (*Figure 7b–e*), the proportion of blood cells released via this technique that could be labelled using the split GAL4 system was relatively low (*Figure 7f*). This suggests that subpopulation cells may favour association with tissues, leaving fewer available to circulate within the hemolymph.

Overall, the distinct dynamics of subpopulation cells in pupal and adult stages (*Figure 6* and *Figure 7*) strongly suggests that these subpopulations are at least partially distinct from each other and highlights their plasticity during development, with their presence, disappearance (via changes in expression shifting them into distinct cell states) and return correlating with changes in the biology of blood cells over the entire lifecourse. While no obvious staining was detected in the lymph gland during larval

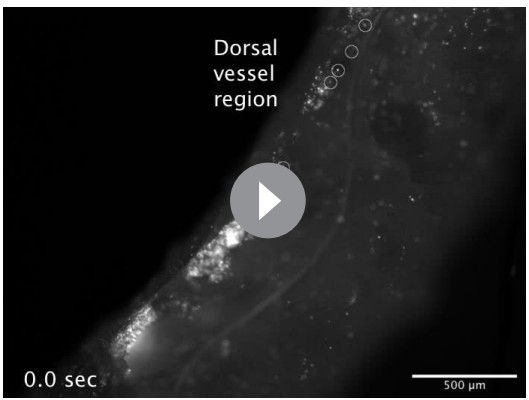

**Video 3.** Dorsal vessel-associated cells can be labelled via *srp*-based split GAL4 reporters in the L3 larva. Movie showing rhythmic movements of cells labelled using *srpHemo-AD* in combination with *srpHemo-DBD* to drive expression from *UAS-stinger* in an L3 larva. The area indicated shows cells on dorsal midline (likely to be nephrocytes) that move in time with pumping of the dorsal vessel (see also *Video 4*, *Video 5* and *Video 7*). These cells can also be seen using *VT32897*-based enhancers. Scale bars represent 500 µm. See *Supplementary file 1* for genotype in full.
https://elifesciences.org/articles/58686#video3

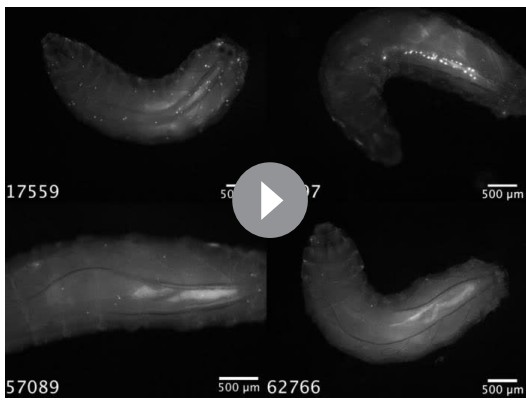

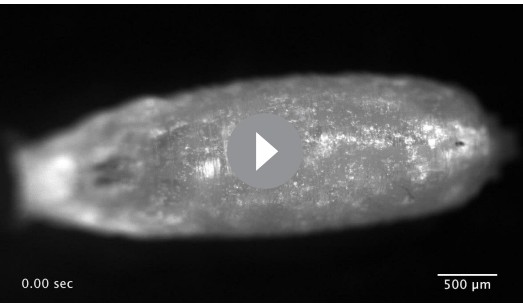

**Video 4.** Localisation and dynamics of subpopulation cells within L3 larvae. Movies showing localisation and movement of cells labelled using split GAL4 lines (*srpHemo-AD* in combination with *VT17559-DBD*, *VT32897-DBD*, *VT57089-DBD*, or *VT62766-DBD*) to drive expression from *UAS-stinger* in L3 larvae. Images taken from timelapse series of single focal planes to enable rapid imaging. Far fewer cells are visible compared to L1 and L2 larvae and the majority of cells detected appear attached to the body wall, since they do not shift their relative positions during larval movements. Movies repeat with second repetition showing examples of rare cells in circulation (illustrated by overlaid tracks). Scale bars represent 500 μm. See *Supplementary file 1* for full list of genotypes.
https://elifesciences.org/articles/58686#video4

**Video 5.** Flow of *srp*-positive cells in circulation within a white pre-pupa. Movie showing movements of *srp*-positive cells within the hemolymph of a white pre-pupa. Cells labelled via *UAS-stinger* expression driven by *srpHemo-AD* in combination with *srpHemo-DBD*. Scale bar represents 500 μm. See *Supplementary file 1* for genotype in full.
https://elifesciences.org/articles/58686#video5

stages using the split GAL4 lines (*Figure 4c–f*), additional lineage-tracing analyses would be required to uncover whether cells derived from the lymph gland contribute to subpopulation cell numbers in pupae or adults.

## Subpopulation cells behave in a functionally distinct manner compared to the overall plasmatocyte population

Given that the VT lines identified above are specifically and dynamically expressed in subpopulations of hemocytes during *Drosophila* development, we next set out to investigate whether the labelled subpopulations are also functionally distinct using a range of immune-relevant assays. The ability of vertebrate macrophages to respond to pro-inflammatory stimuli, such as injuries, can vary according to their activation status (*Arnold et al., 2007*; *Dal-Secco et al., 2015*). To investigate this in our system, a well-established assay of inflammatory migration (*Stramer et al., 2005*) was employed (*Figure 1a*; *Video 1*). Strikingly, following laser-induced wounding, cells labelled by three *VT-GAL4* lines (*VT17559-GAL4*, *VT32897-GAL4* and *VT62766-GAL4*) showed a significantly more potent migratory response to injury. In each case, a greater proportion of labelled subpopulation cells migrated to wounds, compared to the overall hemocyte population as labelled by a pan-plasmatocyte driver (*Figure 8a–c*). Consistent with our results above, plasmatocytes labelled by the VT lines represent a subset of the total number of hemocytes present ventrally in stage 15 embryos (*Figure 8d*).

We next investigated in vivo migration speeds of the embryonic plasmatocyte subpopulations (as per *Figure 1c–d*). Stage 15 embryos were imaged for 1 hr and individual plasmatocyte movements were tracked (*Figure 8e–f*). Only the *VT17559-GAL4*-labelled plasmatocyte subpopulation displayed statistically significantly faster rates of migration compared to the overall plasmatocyte population (labelled using *srpHemo-GAL4*; *Figure 8g*). There were no differences in directionality (cell displacement divided by total path length) for any of the subpopulations, suggesting that the mode of migration was similar across these lines and with that of the overall population (*Figure 8h*).

Apoptotic cell clearance (efferocytosis) represents another evolutionarily-conserved function performed by embryonic plasmatocytes (*Figure 1e*). Therefore, we investigated this function in subpopulations, using numbers of phagosomal vesicles per cell as a proxy for this process (*Evans et al., 2013*). Cells labelled via *VT17559-GAL4*, *VT57089-GAL4* and *VT62766-GAL4* (but not *VT32897-GAL4*) contained fewer phagosomes than the overall plasmatocyte population (*Figure 8i–k*), suggesting that these discrete populations of cells are less effective at removing apoptotic cells inside

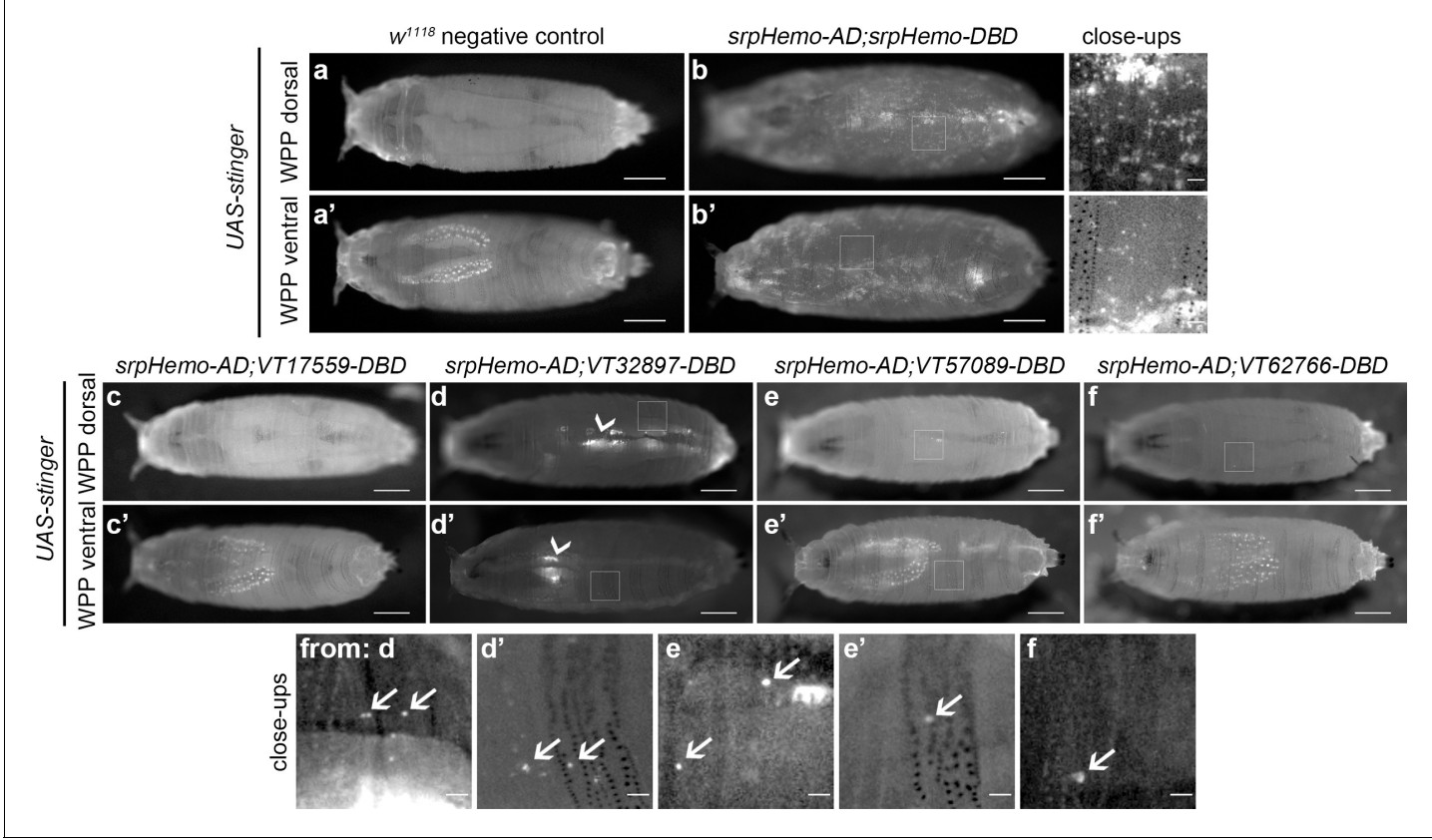

**Figure 5.** Plasmatocyte subpopulations are sparse in white pre-pupae. (**a–b**) Dorsal and ventral views of negative control (**a**, *UAS-stinger*, but no driver) and positive control (**b**, *UAS-stinger* driven by *srpHemo-AD;srpHemo-DBD*) white pre-pupae (WPP); boxes indicate regions shown in close-up views of positive controls. (**c–f**) dorsal and ventral views of WPP containing cells labelled using *srpHemo-AD* and the indicated *VT-DBD* to drive expression from *UAS-stinger*. Very few VT enhancer-labelled cells can be detected in WPP: boxes mark regions shown in close-up views with example hemocytes indicated with an arrow; dorsal vessel-associated and proventricular region/Garland cells can also be observed in *VT32897* WPP (arrowheads in **d** and **d'**, respectively); scale bars represent 500 μm (WPP) or 50 μm (close-ups); WPP images contrast enhanced to 0.3% saturation; close-up images contrast enhanced individually. See *Supplementary file 1* for full list of genotypes.

the developing embryo. To confirm this result dynamically, GFP-myc-2xFYVE, a phosphatidylinositol-3-phosphate reporter (*Wucherpfennig et al., 2003*) was used to measure the rate of phagocytosis in subpopulation cells. This reporter rapidly and transiently localises on the surface of engulfed phagosomes in plasmatocytes (*Roddie et al., 2019*). All subpopulations exhibited lower rates of phagocytosis compared to the overall plasmatocyte population using this reporter (*Figure 8I*), suggesting the differences in numbers of phagosomes per cell result from distinct phagocytic abilities.

Finally, we examined cell size and the shape of labelled plasmatocyte subpopulations. Vertebrate macrophages are highly heterogeneous, with distinct morphologies dependent upon their tissue of residence or polarisation status (*McWhorter et al., 2013*; *Ploeger et al., 2013*; *Rostam et al., 2017*). We found no obvious size or shape differences between *VT-GAL4*-labelled cells and the overall plasmatocyte population (*Figure 8—figure supplement 1a–e*). This was also the case when *VT-GAL4*-positive cells were compared to internal controls (*VT-GAL4*-negative cells within the same embryos) for a range of shape descriptors (*Figure 8—figure supplement 1f–i*). Similarly, we were unable to detect differences in ROS levels (*Figure 8—figure supplement 2*) or the proportion of *VT-GAL4*-labelled plasmatocytes that phagocytosed pHrodo-labelled *E. coli* compared to controls (*Figure 8—figure supplement 3*), two processes associated with pro-inflammatory activation of macrophages (*Benoit et al., 2008*).

Taken together these data show that the subpopulations of plasmatocytes identified via the *VT-GAL4* reporters exhibit functional differences compared to the overall plasmatocyte population (*Table 1*). Therefore, as well as displaying molecular differences in the form of differential enhancer

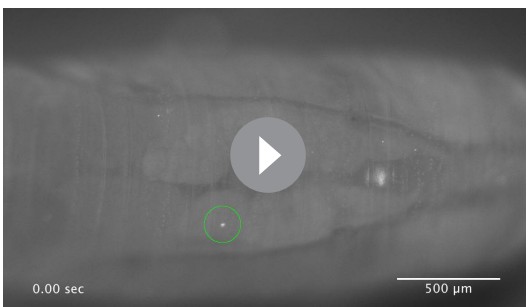

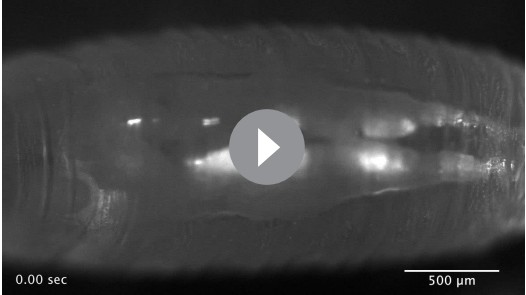

**Video 6.** Movement of *VT57089* subpopulation cells within a white pre-pupa. Movie showing movements of *VT57089* cells within the hemolymph of a white pre-pupa (examples highlighted with green circles). Cells labelled via *UAS-stinger* expression using *srpHemo-AD* and *VT57089-DBD*. Movie plays twice with an overlay of the tracks of cells in circulation shown in repeat. Scale bar represents 500 µm. See *Supplementary file 1* for genotype in full.
https://elifesciences.org/articles/58686#video6

**Video 7.** Movement of *VT32897*-labelled, dorsal vessel-associated, non-hemocyte cells within a white pre-pupa. Movie showing rhythmic movements of cells in a white pre-pupa labelled using *srpHemo-AD* in combination with *VT32897-DBD* to drive expression from *UAS-stinger*. Cells on dorsal midline (likely to be nephrocytes) move in time with pumping of the dorsal vessel in a white pre-pupa. Scale bar represents 500 µm. See *Supplementary file 1* for genotype in full.
https://elifesciences.org/articles/58686#video7

activity, and hence reporter expression, these discrete populations of cells behave differently. This strongly suggests that these cells represent functionally distinct subpopulations and that the plasmatocyte lineage is not homogeneous. Furthermore, not all subpopulations displayed identical functional characteristics, suggesting that there are multiple distinct subtypes present in vivo, although some overlap between subpopulations seems likely. For example, *VT17559-GAL4*-labelled cells were more effective at responding to wounds and migrated more rapidly but carried out less phagocytosis of apoptotic cells. By contrast, *VT32987-GAL4*-labelled cells only displayed improved wound responses (*Figure 8*).

## VT enhancers identify functionally active genes within plasmatocytes

In the original study that analysed the *VT-GAL4* collection, the majority of active enhancer fragments tested were found to control transcription of neighbouring genes (*Kvon et al., 2014*). Thus, genes proximal to enhancers that label plasmatocyte subpopulations represent candidate regulators of immune cell function (*Table 2*; *Figure 9a*). *VT62766-GAL4* labels a subpopulation of plasmatocytes with enhanced migratory responses to injury (*Figure 8a–c*) and this enhancer region is found within the genomic interval containing *paralytic* (*para*), which encodes a subunit of a voltage-gated sodium channel (*Lin et al., 2009*), and upstream of the 3′ end of *Calnexin14D* (*Cnx14D*; *Figure 9a*). *Cnx14D* encodes a calcium-binding chaperone protein resident in the endoplasmic reticulum (*Christodoulou et al., 1997*). Alterations in calcium dynamics are associated with clearance of apoptotic cells (*Cuttell et al., 2008*; *Gronski et al., 2009*) and modulating calcium signalling within plasmatocytes alters their ability to respond to wounds (*Weavers et al., 2016*). Therefore, given the association of *Cnx14D* with the *VT62766* enhancer and the potential for plasmatocyte behaviours to be modulated by altered calcium dynamics, we examined whether misexpressing *Cnx14D* in all plasmatocytes was sufficient to cause these cells to behave more similarly to the *VT62766* subpopulation. Critically, pan-hemocyte expression of *Cnx14D* stimulated wound responses with elevated numbers of plasmatocytes responding to injury compared to controls (*Figure 9b–c*), consistent with the enhanced wound responses of the endogenous *VT62766-GAL4*-positive plasmatocyte subpopulation (*Figure 8c*). This reveals that genes proximal to subpopulation-defining enhancers represent candidate genes in dictating the biology of cells in those subpopulations. More importantly, misexpression of a subpopulation-linked gene promotes a similar behaviour to that subpopulation in the wider plasmatocyte population.

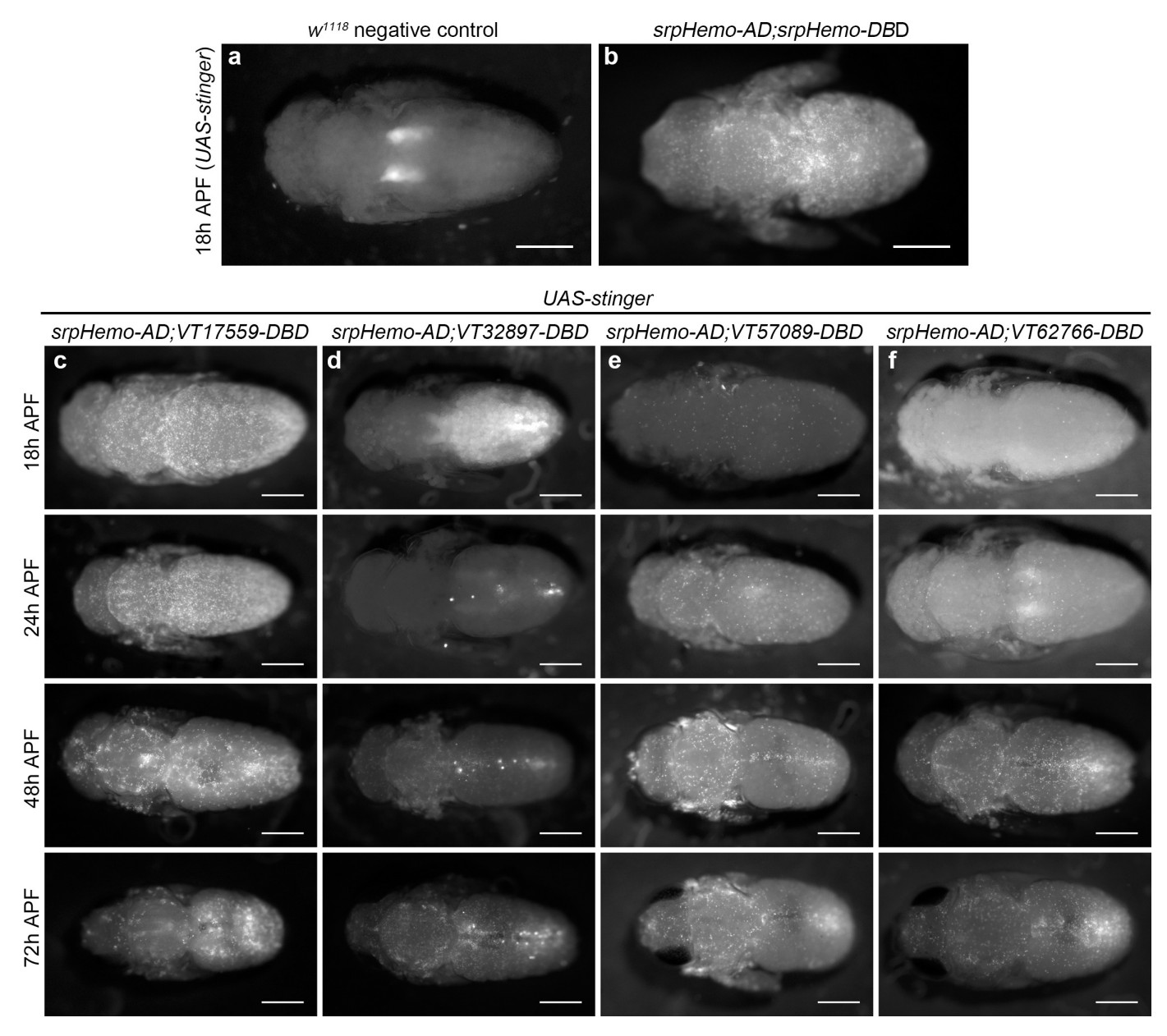

**Figure 6.** Plasmatocyte subpopulations return with distinct dynamics during pupal development. (**a–b**) Dorsal images of negative control (**a**, no GAL4) and positive control pupae (**b**, labelled via *srpHemo-AD;srpHemo-DBD*) at 18 hr after puparium formation (APF). (**c–f**) dorsal images showing localisation of cells labelled using *srpHemo-AD* and *VT-DBD* (VT enhancers used to drive *DBD* expression indicated above panels) to drive expression of *UAS-stinger* during pupal development from 18 hr AFP to 72 hr APF. All image panels contrast enhanced to 0.3% saturation to reveal localisation of labelled cells due to differing intensities of reporter line expression. Scale bars represent 500 µm. See *Supplementary file 1* for full list of genotypes.

## Plasmatocyte subpopulations can be modulated via exposure to enhanced levels of apoptosis

Having defined functional differences in embryonic plasmatocyte subpopulations and characterised how these populations shift during development and ageing, we sought to identify the processes via which these subpopulations were specified. In vertebrates, a range of stimuli drive macrophage heterogeneity and polarisation (*Martinez and Gordon, 2014*; *Murray, 2017*), with apoptotic cells able to polarise macrophages towards anti-inflammatory phenotypes (*A-Gonzalez et al., 2017*; *de Oliveira Fulco et al., 2014*). In the developing fly embryo, high apoptotic cell burdens impair

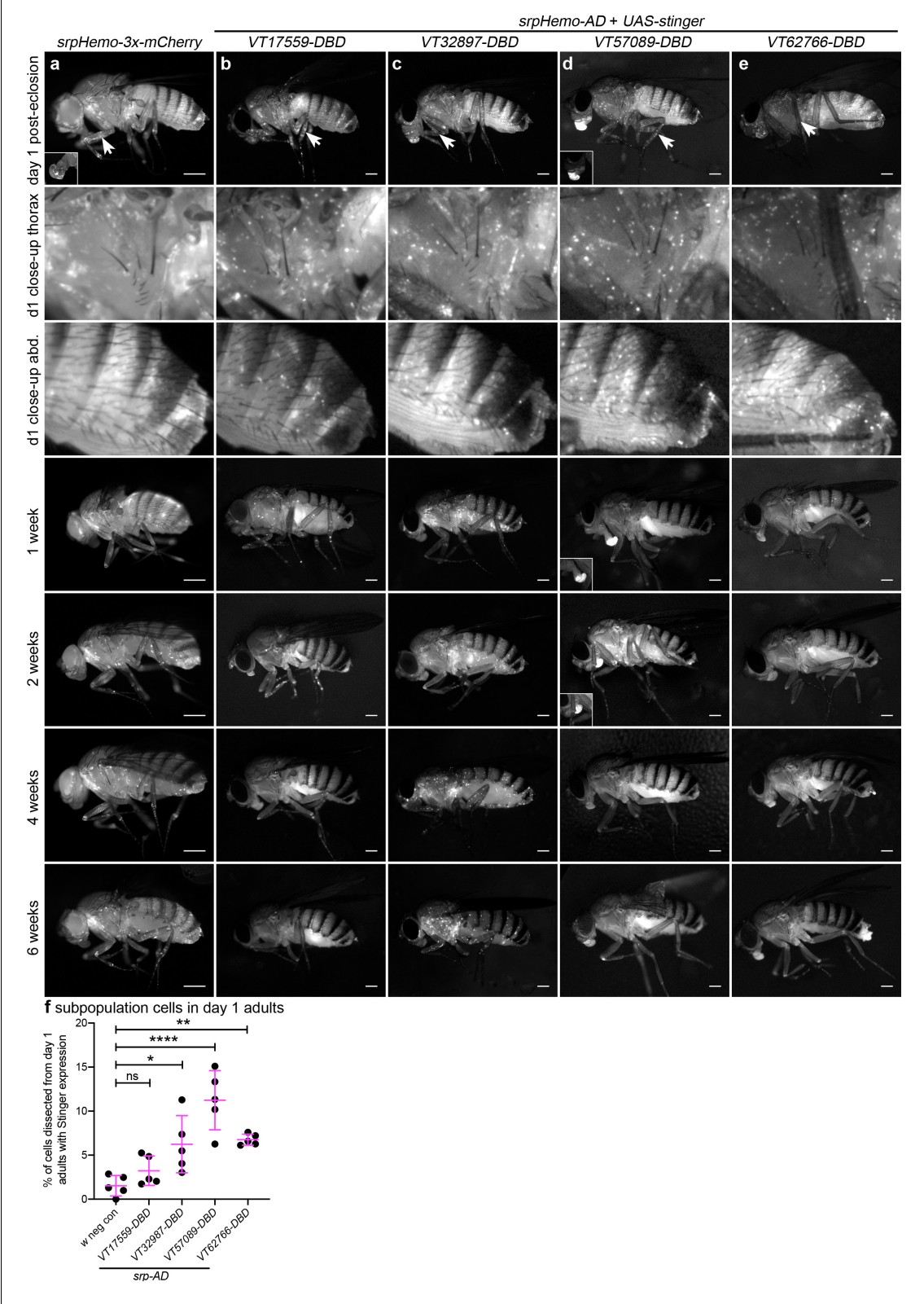

**Figure 7.** Plasmatocyte subpopulations exhibit distinct localisations and dynamics as adults age. (a–e) Representative lateral images of adult flies between 0 and 6 weeks of age showing localisation of cells labelled using *srpHemo-3x-mCherry* (a, positive control), or split GAL4 to drive expression of stinger (b-e, *srpHemo-AD;VT-DBD*). The VT enhancers used to drive expression of the DNA-binding domain (*DBD*) of *GAL4* correspond to *VT17559* (b), *VT32897* (c), *VT57089* (d), and *VT62766* (e); inset images show alternative view of proboscis region from same fly (a) or at a reduced level of

*Figure 7 continued on next page*

*Figure 7 continued*

brightness to reveal cellular detail (d). Images contrast enhanced to 0.15% saturation (a–c, e) or 0.75% (d) to reveal localisation of labelled cells due to differing intensities of reporter line expression. Arrows in top row indicate hemocytes in the legs; 2nd and 3rd rows show close-up of thorax and abdomen of day one flies; at least five flies were imaged for each timepoint; scale bars represent 500 µm. (f) Scatterplot showing proportion of cells dissected from day one adults that were labelled using *srpHemo-AD* and the *VT-DBD* transgenes indicated to drive expression from *UAS-stinger*. One-way ANOVA used to compare to negative control flies ($w^{1118}$;*UAS-stinger/+*) with split GAL4 VT lines: n = 5 dissections per genotype; p=0.60 (*VT17559*), p=0.013 (*VT32897*), p<0.0001 (*VT57089*), and p=0.0063 (*VT62766*). Lines and error bars represent mean and standard deviation, respectively; ns, *, ** and **** denote not significant (p>0.05), p<0.05, p<0.01, and p<0.0001, respectively. See *Supplementary file 1* for full list of genotypes. The online version of this article includes the following source data for figure 7:

**Source data 1.** Numerical data used to plot panel (f) of *Figure 7*.

wound responses (*Armitage et al., 2020*; *Roddie et al., 2019*), consistent with reprogramming of plasmatocytes towards less wound-responsive states. In order to test whether apoptotic cells might regulate plasmatocyte subpopulations, we exposed plasmatocytes to increased levels of apoptosis in vivo. In the developing fly embryo, both glial cells and plasmatocytes contribute to the clearance of apoptotic cells. We, and others, have previously shown that loss of *repo*, a transcription factor required for glial specification (*Campbell et al., 1994*; *Halter et al., 1995*; *Xiong et al., 1994*), leads to decreased apoptotic cell clearance by glia (*Shklyar et al., 2014*), and a subsequent challenge of plasmatocytes with increased levels of developmental apoptosis (*Figure 10a–b*; *Armitage et al., 2020*). Therefore, a *repo* mutant background represents an established model with which to stimulate plasmatocytes with enhanced levels of apoptosis.

Using *srpHemo-H2A-mCherry* to mark all plasmatocytes within the embryo (*Figure 10c*), we quantified the proportion of plasmatocytes labelled via *VT-GAL4* transgenes in *repo* mutants compared to controls (*Figure 10d–h*). Increased exposure to apoptotic death shifted plasmatocytes out of each subpopulation (*Figure 10d–h*). Subpopulations exhibited differing sensitivities to contact with apoptotic cells, with numbers of *VT62766-GAL4*-labelled cells undergoing the largest decrease in a *repo* mutant background (*Figure 10h*). These results therefore reveal a mechanism via which the molecularly and functionally distinct subpopulations of plasmatocytes we have identified can be manipulated using an evolutionarily conserved, physiological stimulus (apoptotic cells) relevant to immune cell programming.

## Discussion

We have identified molecularly and functionally distinct subpopulations of *Drosophila* macrophages (plasmatocytes). These subpopulations showed functional differences compared to the overall plasmatocyte population, exhibiting enhanced responses to injury, faster migration rates and reduced rates of apoptotic cell clearance within the developing embryo. These subpopulations are highly plastic with their numbers varying across development, in line with the changing behaviours of *Drosophila* blood cells across the lifecourse. That these discrete populations of plasmatocytes represent bona fide subpopulations is evidenced by the finding that numbers of cells within subpopulations can be manipulated via exposure to enhanced levels of apoptotic cell death in vivo. Furthermore, pan-hemocyte expression of a gene (*Cnx14D*) linked to one of the enhancers used to visualise these subpopulations (*VT62766-GAL4*) shifts the behaviour of these cells towards a more wound-responsive state, resembling the behaviour of *VT62766-GAL4*-labelled cells. Taken together this data strongly suggests that *Drosophila* blood cell lineages are more complex than previously known.

Vertebrate macrophage lineages show considerable heterogeneity due to the presence of circulating monocytes, a wide variety of tissue resident macrophages and a spectrum of activation states that can be achieved (*Gordon and Plüddemann, 2017*; *Wynn et al., 2013*). Whether simpler organisms such as *Drosophila* exhibit heterogeneity within their macrophage-like lineages has been a topic of much discussion and hints in the literature suggest this as a possibility. Braun and colleagues identified variation in reporter expression within plasmatocytes in an enhancer trap screen, but without associating these with functional differences (*Braun et al., 1997*), while heterogeneity has also been suggested previously (*Anderl et al., 2016*; *Kurucz et al., 2007a*). For instance, non-uniform expression has been reported for plasmatocyte genes such as *hemolectin* (*Goto et al., 2003*), *hemese*, *nimrod* (*Kurucz et al., 2007b*; *Kurucz et al., 2007a*), *croquemort*, TGF-β family members

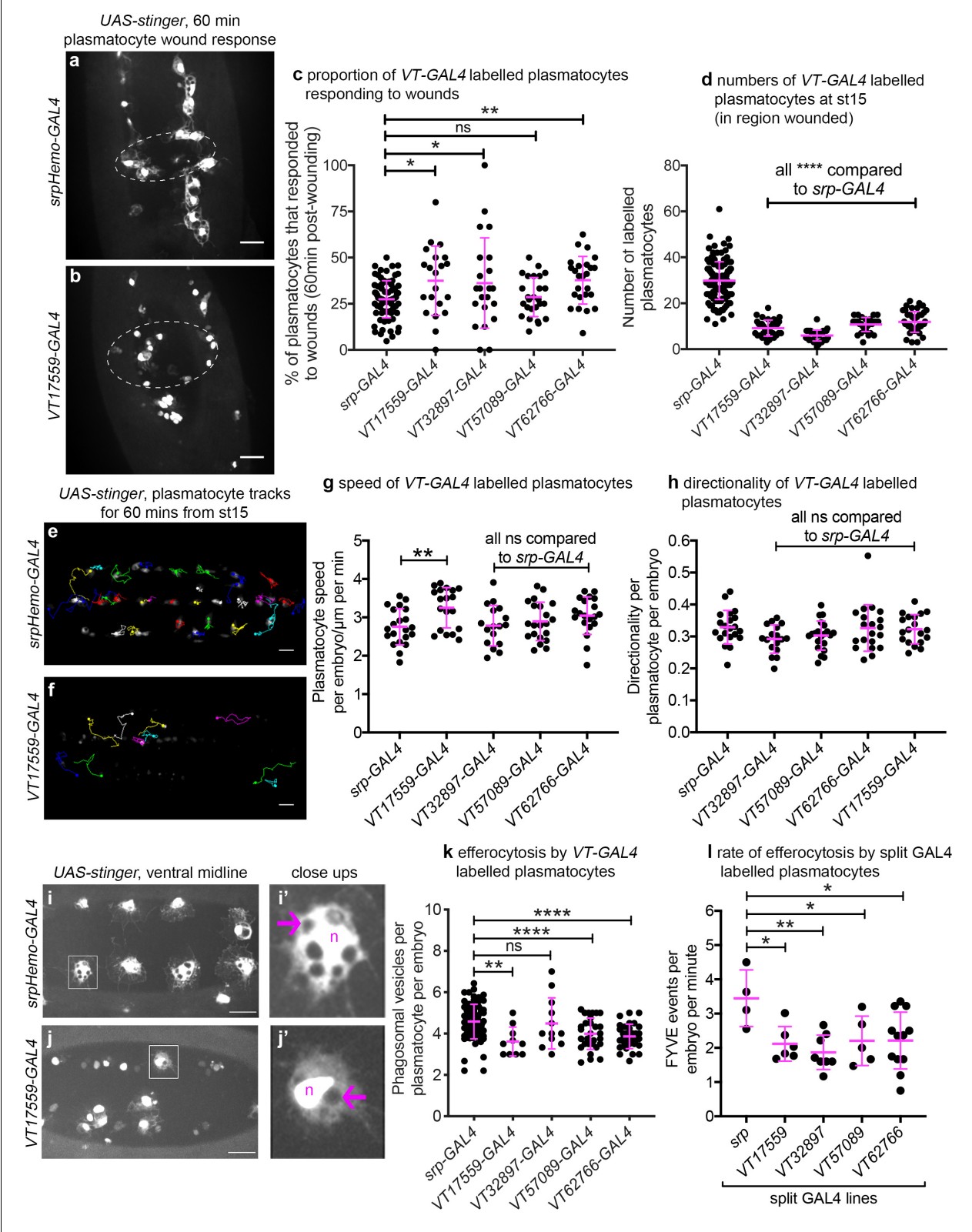

**Figure 8.** *Drosophila* plasmatocyte subpopulations demonstrate functional differences compared to the overall plasmatocyte population. (**a–b**) Example images showing plasmatocyte wound responses at 60 min post-wounding (maximum projections of 15 μm deep regions). Cells labelled via *UAS-stinger* using *srpHemo-GAL4* (**a**) and *VT17559-GAL4* (**b**); dotted lines show wound edges. (**c–d**) Scatterplots showing percentage of *srpHemo-GAL4* (control) or *VT-GAL4*-labelled plasmatocytes responding to wounds at 60 min (**c**) or total numbers of labelled plasmatocytes in wounded region (**d**); *Figure 8 continued on next page*

*Figure 8 continued*

p=0.018, 0.041, 0.99, 0.0075 compared to *srpHemo-GAL4* (n = 77, 21, 22, 26, 25) (c); p<0.0001 compared to *srpHemo-GAL4* for all lines (n = 139, 35, 37, 30, 44) (d). (e–f) Example tracks of plasmatocytes labelled with GFP via *srpHemo-GAL4* (e) and *VT17559-GAL4* (f) during random migration on the ventral side of the embryo for 1 hr at stage 15. (g–h) Scatterplots showing speed per plasmatocyte, per embryo (g) and directionality (h) at stage 15 in embryos containing cells labelled via *srpHemo-GAL4* (control) or the *VT-GAL4* lines indicated; p=0.0097, 0.999, 0.82, 0.226 compared to *srpHemo-GAL4* (n = 21, 19, 17, 21, 20) (g); p=0.998, 0.216, 0.480, 0.999 compared to *srpHemo-GAL4* (n = 21, 19, 17, 21, 20) (h). (i–j) Example images of cells on the ventral midline at stage 15 with labelling via *UAS-stinger* expression using *srpHemo-GAL4* (i) and *VT17559-GAL4* (j); plasmatocytes shown in close-up images (i', j') are indicated by white boxes in main panels; arrows show phagosomal vesicles, 'n' marks nucleus; n.b. panels contrast enhanced independently to show plasmatocyte morphology. (k) Scatterplot showing phagosomal vesicles per plasmatocyte, per embryo at stage 15 (measure of efferocytosis/apoptotic cell clearance); cells labelled via *srpHemo-GAL4* (control) or the *VT-GAL4* lines indicated; p=0.0020, 0.99, 0.0040, 0.0002 compared to *srpHemo-GAL4* (n = 76, 10, 12, 29, 31). (l) Scatterplot showing number of times 2x-FYVE-EGFP sensor recruited to phagosomes (FYVE events) per plasmatocyte, per embryo in plasmatocytes labelled via the split GAL4 system; p=0.019, 0.0034, 0.039 and 0.015 compared to *srp* control (n = 4, 6, 8, 5 and 12 embryos). Lines and error bars represent mean and standard deviation, respectively (all scatterplots); one-way ANOVA with a Dunnett's multiple comparison test used to compare VT lines with *srp* controls in all datasets; ns, *, **, and **** denote not significant (p>0.05), p<0.05, p<0.01, and p<0.0001, respectively. All scale bars represent 20 µm. See *Supplementary file 1* for full list of genotypes. N.b. *Figure 8—figure supplements 1–3* show analysis of subpopulation cell morphology, ROS levels and phagocytosis in response to immune challenge, respectively.

The online version of this article includes the following source data and figure supplement(s) for figure 8:

**Source data 1.** Numerical data used to plot panel (c) of *Figure 8*.
**Source data 2.** Numerical data used to plot panel (d) of *Figure 8*.
**Source data 3.** Numerical data used to plot panel (g) of *Figure 8*.
**Source data 4.** Numerical data used to plot panel (h) of *Figure 8*.
**Source data 5.** Numerical data used to plot panel (k) of *Figure 8*.
**Source data 6.** Numerical data used to plot panel (l) of *Figure 8*.
**Figure supplement 1.** *VT-GAL4*-labelled subpopulations show no gross differences in morphology compared to non-labelled plasmatocytes.
**Figure supplement 1—source data 1.** Numerical data used to plot panel (f) of *Figure 8—figure supplement 1*.
**Figure supplement 1—source data 2.** Numerical data used to plot panel (g) of *Figure 8—figure supplement 1*.
**Figure supplement 1—source data 3.** Numerical data used to plot panel (h) of *Figure 8—figure supplement 1*.
**Figure supplement 1—source data 4.** Numerical data used to plot panel (i) of *Figure 8—figure supplement 1*.
**Figure supplement 2.** *VT-GAL4*-labelled plasmatocytes show no gross differences in their ROS levels compared to the overall population.
**Figure supplement 2—source data 1.** Numerical data used to plot panel (f) of *Figure 8—figure supplement 2*.
**Figure supplement 3.** *VT-GAL4*-labelled plasmatocytes show no gross differences in their phagocytosis of *E. coli* compared to the overall population.
**Figure supplement 3—source data 1.** Numerical data used to plot panel (c) of *Figure 8—figure supplement 3*.

**Table 1.** Summary of plasmatocyte subpopulation characteristics and their developmental regulation.

| Subpopulation | Subpopulation characteristics (compared to overall population): | | | | | | Subpopulations in: | | | | |
| | Wound responses | Migration speed | Efferocytosis | ROS levels | Phagocytosis of *E. coli* | | Embryos | Larvae | Pupae | Newly hatched adults | Aged adults |
|---|---|---|---|---|---|---|---|---|---|---|---|
| VT17559 | ↓ | ↓ | ↓ | no difference | no difference | | distinct subpopulation | very few cells labelled | large numbers labelled by 18 hr APF | large numbers present | largely absent by 2 weeks |
| VT32897 | ↓ | no difference | only decreased in FYVE | no difference | no difference | | distinct subpopulation (fewest cells) | few cells labelled + nephrocytes and garland cells (?) | large numbers labelled by 72 hr APF | large numbers present | labelled cells persist |
| VT57089 | no difference | no difference | ↓ | no difference | no difference | | distinct subpopulation | almost no cells labelled + Bolwig Organ (?) | steady increase in numbers labelled | large numbers present | largely absent by 1 week |
| VT62766 | ↓ | no difference | ↓ | no difference | no difference | | distinct subpopulation | almost no cells labelled | large numbers labelled by 48 hr APF | large numbers present | largely absent by 1 week |

The online version of this article includes the following source data for Table 1:
Source data 1. Source data for *Table 1*.Summary of plasmatocyte subpopulation characteristics and their developmental regulation.

**Table 2.** VT enhancer region location and neighbouring genes.

| VT enhancer | Genomic region* | Nearest genes | Distance of enhancer from gene |
|---|---|---|---|
| VT17559 | chr2R: 12,069,698–12,070,780 | Lis-1 | overlapping |
| | | CG8441 | 2,929bp upstream |
| | | Ptp52F | 3,887bp downstream |
| VT32897 | chr3L: 18,631,149–18,633,281 | MYPT-75D | overlapping |
| | | bora | 13,299bp downstream |
| | | not | 15,921bp downstream |
| VT57089 | chrX: 4,961,770–4,962,316 | ovo | overlapping |
| | | CG32767 | 3,290bp upstream |
| | | CR44833 | 3,870bp downstream |
| VT62766 | chrX: 16,406,666–16,408,777 | para | overlapping |
| | | Cnx14D | 10,404bp upstream |
| | | CG9903 | 26,520bp upstream |

* *D. melanogaster* Apr. 2006 (BDGP R5/dm3) Assembly.

Data taken from http://enhancers.starklab.org/.

The online version of this article includes the following source data for Table 2:

Source data 1. Source data for *Table 2*.VT enhancer region location and neighbouring genes.

(*Clark et al., 2011*), and the iron transporter *malvolio* (*Folwell et al., 2006*). The ease of extracting larval hemocytes has meant these cells have received more attention than their embryonic counterparts; recent transcriptional profiling approaches via scRNAseq that emerged during preparation of this manuscript have suggested the existence of distinct larval blood cell populations in *Drosophila* (*Cattenoz et al., 2020*; *Fu et al., 2020*; *Tattikota et al., 2019*) and provided further confirmation of the existence of self-renewing/proliferating plasmatocytes at this stage of development (*Makhijani et al., 2011*). Similar approaches have been taken to study the cells of the lymph gland (*Cho et al., 2020*), though further work is required to establish whether the blood cells generated in this second haematopoietic wave contribute to subpopulation numbers in pupae and adults. While it has been suggested that some of these molecular differences may, at least in part, reflect the presence of transient progenitor states (*Tattikota et al., 2019*), these studies identified a number of potentially different functional groups, including more immune-activated cell populations displaying expression signatures reflective of active Toll and JNK signalling (*Cattenoz et al., 2020*; *Fu et al., 2020*; *Tattikota et al., 2019*). Therefore, our identification of developmentally regulated subpopulations, coupled with this recent evidence from larvae, strongly points to functional heterogeneity within the plasmatocyte lineage.

How do the functionally distinct subpopulations we have uncovered relate to the transcriptionally-defined clusters revealed via scRNAseq? These approaches profiled L3 larval hemocytes (*Cattenoz et al., 2020*; *Fu et al., 2020*; *Tattikota et al., 2019*), the stage at which fewest subpopulation cells can be identified. Therefore, it is possible that VT-labelled cells do not correspond to any of the scRNAseq clusters: subpopulation cells in L3 larvae may represent high expressors from earlier in development that are only marked due to perdurance of fluorescent protein. Alternatively, in terms of function, it could be concluded that the VT-labelled subpopulations display a degree of immune activation given their decreased efficiency at removing apoptotic cells and increased responses to wounds. These subpopulations could thus relate to clusters displaying signatures of immune activation (PM3-PM7; *Tattikota et al., 2019* PL-Rel, PL-vir1, PL-AMP; *Cattenoz et al., 2020*). In contrast to clusters predicted to be proliferative (PM9-11/PL-prolif/PL-Inos), these activated clusters did not show a bias in their distribution between circulation and sessile patches (*Cattenoz et al., 2020*; *Tattikota et al., 2019*), similar to the localisation of VT-labelled cells. While we have not categorically identified which genes are regulated by the VT-enhancers that define subpopulations (see *Table 2* for candidates), transcripts of several of these candidates are enriched in PM6, an immune-activated cluster, and PM12, which accounts for less than 1% of plasmatocytes and has been difficult to classify since it is defined by uncharacterised genes (*Tattikota et al., 2019*).

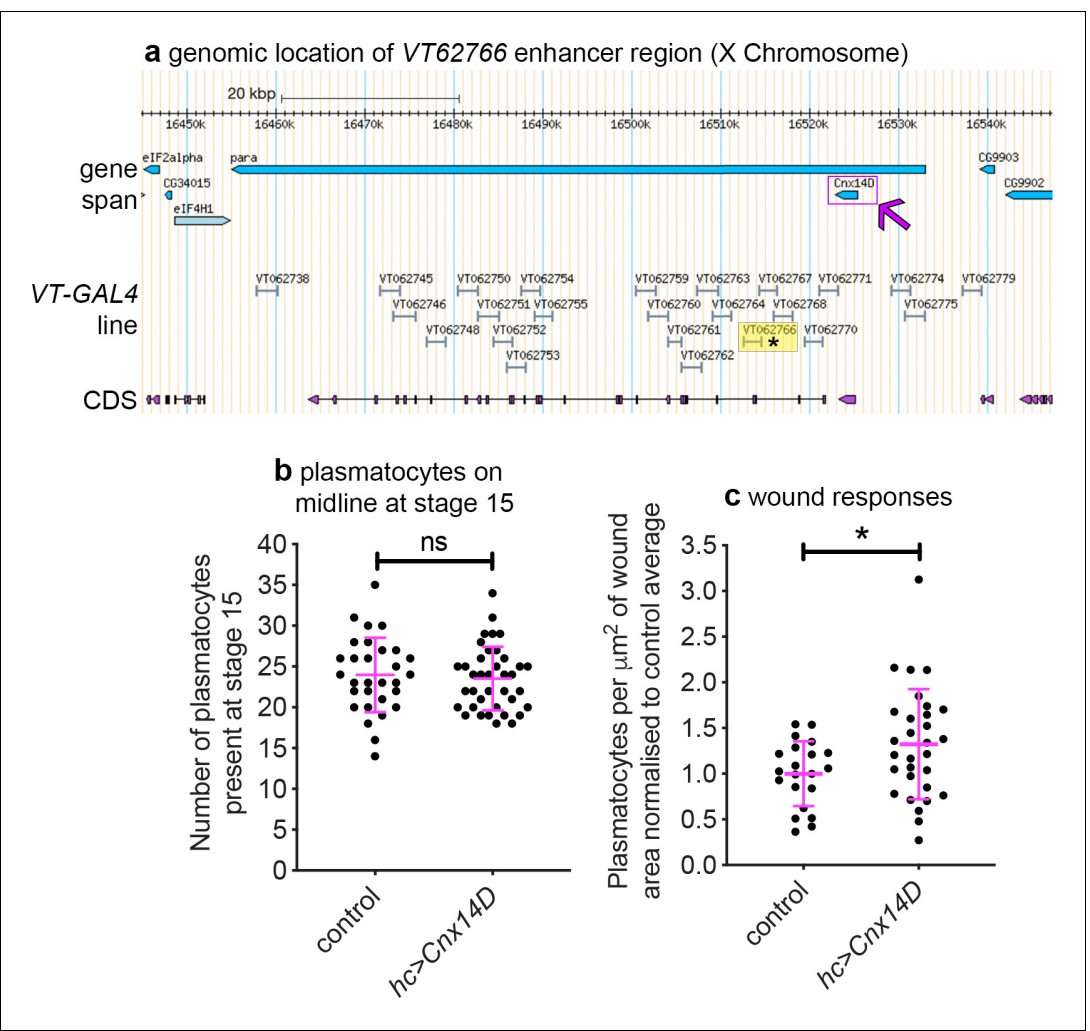

**Figure 9.** Misexpression of *Cnx14D* improves plasmatocyte inflammatory responses to injury. (**a**) Chromosomal location of the *VT62766-GAL4* enhancer region; only one transcript is shown for *para*, which possesses multiple splice variants. The *VT62766* region is highlighted in yellow and by an asterisk; *Cnx14D* (indicated by magenta arrow) lies within *para*. (**b**) Scatterplot showing numbers of plasmatocytes present at stage 15 on the ventral side of the embryo ahead of wounding in controls and on misexpression of *Cnx14D* in all hemocytes using both *srpHemo-GAL4* and *crq-GAL4* (*hc>Cnx14D*); n = 30 and 38 for control and *hc>Cnx14D* embryos, respectively, p=0.670 via Student's t-test. (**c**) Scatterplot of wound responses 60 min post-wounding (number of plasmatocytes at wound, normalised for wound area and to control responses); n = 21 and 30 for control and *hc>Cnx14D* embryos, respectively; p=0.0328 via Student's t-test. Line and error bars represent mean and standard deviation, respectively (**b–c**). See *Supplementary file 1* for full list of genotypes.

The online version of this article includes the following source data for figure 9:

**Source data 1.** Numerical data used to plot panel (**b**) of *Figure 9*.
**Source data 2.** Numerical data used to plot panel (**c**) of *Figure 9*.

While the field is still at an early stage, further characterisation and new tools based on marker genes will enable these different clusters and subpopulations to be more carefully compared.

The subpopulations we have identified are significantly reduced in L3 larvae and consequently may represent functional heterogeneity more relevant to other developmental stages. It is clear that the biology of *Drosophila* blood cells varies significantly across the lifecourse: for instance plasmatocytes play strikingly different functional roles in embryos and larvae (*Charroux and Royet, 2009*; *Defaye et al., 2009*), shifting from developmental roles to host defence. Additionally, plasmatocytes undergo directed migration to sites of injury in embryos and pupae (*Moreira et al., 2011*;

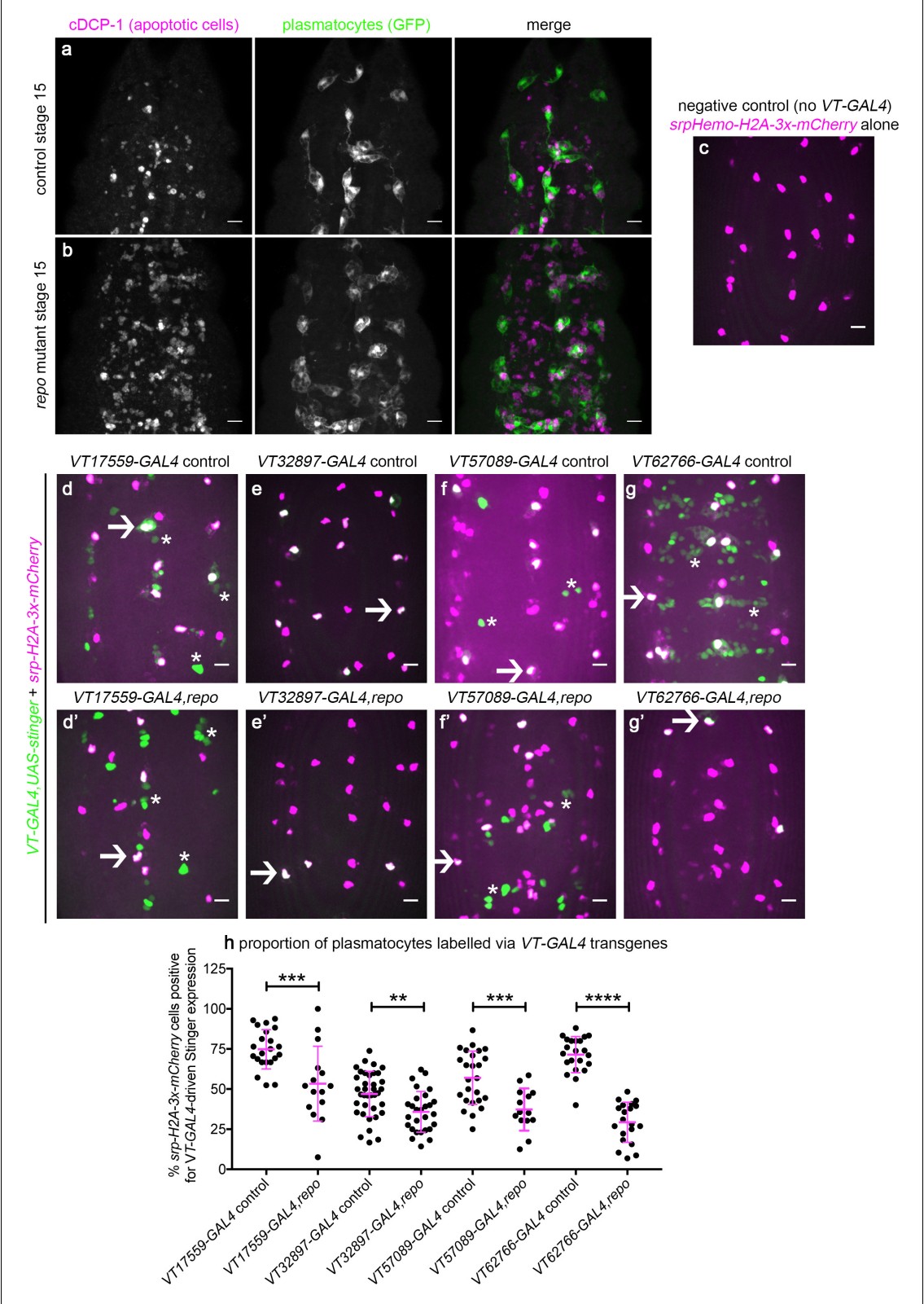

**Figure 10.** *Drosophila* plasmatocyte subpopulation identity can be controlled through exposure to apoptotic cells. (**a–b**) Maximum projections showing apoptotic cells (via anti-cDCP-1 staining, magenta in merge) and plasmatocytes (via anti-GFP staining, green in merge) at stage 15 on the ventral midline in control (**a**) and *repo* mutant embryos (**b**). (**c–g**) maximum projections of the ventral midline showing a negative control embryo (**c**) and embryos containing *VT-GAL4*-labelled plasmatocytes at stage 15 in control (**d–g**) and *repo* mutant embryos (**d'–g'**). *VT-GAL4* used to drive *UAS-stinger*

*Figure 10 continued on next page*

*Figure 10 continued*

expression (green) and *srpHemo-H2A-3x-mCherry* used to label plasmatocytes (magenta). Arrows and asterisks indicate examples of *VT-GAL4*-positive plasmatocytes and non-plasmatocyte cells, respectively; note loss of non-plasmatocyte *VT-GAL4* expression in *repo* mutants versus controls for *VT62766-GAL4*. (h) Scatterplot showing percentage of H2A-3x-mCherry-positive cells that are also positive for *VT-GAL4* driven Stinger expression in control and *repo* mutant embryos at stage 15. Student's t-test used to show significant difference between controls and *repo* mutants (p=0.0009, n = 22, 15 for *VT17559-GAL4* lines; p=0.0017, n = 37, 28 for *VT32897-GAL4* lines; p=0.0005, n = 25, 14 for *VT57089-GAL4* lines; p<0.0001, n = 22, 20 for *VT62766-GAL4* lines). Scale bars represent 10 μm (a–g); lines and error bars represent mean and standard deviation (h); **, ***, and **** denote p<0.01, p<0.001, and p<0.0001, respectively. See *Supplementary file 1* for full list of genotypes.

The online version of this article includes the following source data for figure 10:

**Source data 1.** Numerical data used to plot panel (h) of *Figure 10*.

---

*Stramer et al., 2005*), stages of development when subpopulation cells are most obvious. In contrast, hemocytes are captured from circulation via adhesion in L3 larvae and their migratory abilities are less obvious (*Babcock et al., 2008*). These functional differences are reflected in molecular differences between embryonic and larval blood cells revealed via bulk RNAseq (*Cattenoz et al., 2020*), with reprogramming to suit the different requirements of these cells within larvae (*Charroux and Royet, 2009*; *Defaye et al., 2009*), potentially explaining why our VT enhancer-labelled subpopulations are substantially decreased at that stage. Transcriptional changes are also associated with steroid hormone-mediated signalling in pupae (*Regan et al., 2013*) and this hormone (ecdysone) can also drive alterations in blood cell behaviours (*Sampson et al., 2013*). Thus, steroid hormone signalling represents a potential candidate mechanism to drive re-emergence of subpopulations in time for metamorphosis.

In higher vertebrates, erythro-myeloid precursor/progenitor cells seed the developing embryo to give rise to tissue resident macrophage populations (*Gomez Perdiguero et al., 2015*; *Hoeffel and Ginhoux, 2018*; *Mass et al., 2016*). Intriguingly, the localisation of subpopulations in larvae and adult flies shows some biases between subpopulation lines and the overall population, hinting at the potential for some degree of tissue residency in *Drosophila* or that individual tissues and their microenvironments can imprint tissue-specific transcriptional programmes upon plasmatocytes in those locations. Hemocytes are known to localise to and/or play specialised roles at a range of tissues including the respiratory epithelia (*Sanchez Bosch et al., 2019*), dorsal vessel (*Cevik et al., 2019*), ovaries (*Van De Bor et al., 2015*), wings (*Kiger et al., 2001*), gut (*Ayyaz et al., 2015*), and proventriculus (*Zaidman-Rémy et al., 2012*). It is therefore tempting to speculate that particular subpopulations could be recruited to these locations or differentiate in situ in order to carry out specific functions. As hemocytes are thought to be relatively immobile in larvae and adult flies (*Makhijani et al., 2011*; *Sanchez Bosch et al., 2019*), recruitment may occur during embryonic stages or in pupae when these cells are more motile (*Moreira et al., 2011*; *Paladi and Tepass, 2004*). Vertebrate studies typically show acquisition of tissue resident transcriptional profiles after homing (*Gosselin et al., 2014*; *Lavin et al., 2014*) – therefore, it seems more likely that the ultimate environment in which plasmatocytes find themselves shapes their transcriptional profile. Further fine-tuning in response to local stimuli, such as via phagocytosis (*A-Gonzalez et al., 2017*), may also play a role in this process, as seen with increased exposure to apoptotic cells reducing plasmatocyte subpopulations in the developing embryo. Future work will establish the extent to which we can use flies to model the mechanisms by which tissue microenvironments sculpt macrophage heterogeneity.

Macrophage diversity enables these important innate immune cells to operate in a variety of niches and carry out a wide variety of functions in vertebrates. Our data demonstrate that not all macrophages are equivalent within the developing *Drosophila* embryo, although the enhancers we have used to identify plasmatocyte subpopulations do not correspond to markers used in defining vertebrate macrophage polarisation or tissue resident populations in an obvious way. Therefore, how the subpopulations we have uncovered map onto existing vertebrate paradigms remains an open question. Nonetheless, these *Drosophila* subpopulations could be viewed as displaying a pro-inflammatory skewing of immune cell behaviours, given their enhanced wound responses, faster rates of migration and decreased efferocytic capacity. Pro-inflammatory macrophages (M1-like) in vertebrates are associated with clearance of pathogens, release of pro-inflammatory cytokines and, most pertinently, initial responses to injury (*Benoit et al., 2008*). In contrast, anti-inflammatory macrophages (M2-like) are more allied with tissue development and repair (*Krzyszczyk et al., 2018*) and

can display enhanced rates of efferocytosis (*Lingnau et al., 2007*; *Ogden et al., 2005*; *Zizzo et al., 2012*).

Apoptotic cell clearance can promote anti-inflammatory states in vertebrates (*Fadok et al., 1998*). Consequently, it is both consistent and compelling that exposure of *Drosophila* plasmatocytes to excessive levels of apoptotic cells dampens their inflammatory responses to injury and rates of migration in the developing embryo (*Armitage et al., 2020*; *Evans et al., 2013*; *Roddie et al., 2019*) and also shifts cells out of the more wound-responsive and potentially pro-inflammatory subpopulations we have discovered. Previous work suggests that macrophage polarisation may exist in *Drosophila* with infection causing hemocytes to prioritise aerobic glycolysis (*Krejčová et al., 2019*), similar to the situation on acquisition of pro-inflammatory states in vertebrates (*Van den Bossche et al., 2017*). Parallels also exist in the eye following UV-induced damage, with upregulation of the M2 marker arginase in hemocytes as part of repair responses (*Neves et al., 2016*). Furthermore, TGF-β signalling is associated with promotion of anti-inflammatory characteristics in vertebrates during resolution of inflammation (*Fadok et al., 1998*) and these molecules can be found in discrete sets of hemocytes on injury and infection in adult flies (*Clark et al., 2011*). Thus, despite significant evolutionary distance between flies and vertebrates, comparable processes and mechanisms may control the behaviours of their innate immune cells.

We have concentrated on using the VT enhancers as reporters to follow subpopulation behaviour in vivo. While a lack of associated gene expression does not preclude the use of these enhancers to label subpopulations, these elements also potentially identify genes required for specific functions associated with each subpopulation. For instance, the *VT17559* enhancer overlaps *Lisencephaly-1*, which has been shown to be expressed in hemocytes (*Williams, 2009*). Furthermore, misexpression of *Cnx14D*, located proximally to the *VT62766* enhancer, was sufficient to improve overall wound responses, paralleling the behaviour of the *VT62766-GAL4*-labelled subpopulation. Cnx14D can bind calcium and therefore potentially modulates calcium signalling within plasmatocytes. Calcium signalling is known to influence wound responses in flies (*Weavers et al., 2016*) and plays a central role during phagocytosis of apoptotic cells (*Cuttell et al., 2008*; *Gronski et al., 2009*). Therefore, a molecule such as Cnx14D, which also has a known role in phagocytosis in *Dictyostelium* (*Müller-Taubenberger et al., 2001*), could help fine-tune the behaviour of specific macrophage subpopulations. When considered in combination with the ability to manipulate the numbers of cells within subpopulations with physiologically relevant stimuli, the functional linkage of candidate genes with subpopulation behaviours strongly suggests that we have identified bona fide functionally and molecularly distinct macrophage subpopulations in the fly.

In conclusion, we have demonstrated that *Drosophila* macrophages are a heterogeneous population of cells with distinct functional capabilities. We have characterised novel tools with which to visualise these subpopulations and have used these tools to reveal functional differences between these subpopulations and the general complement of hemocytes. Furthermore, we have shown that these subpopulations can be manipulated by exposure to apoptotic cells and can be linked to specific functional players. Therefore, we have further established *Drosophila* as a model for studying macrophage heterogeneity and immune programming and demonstrate that macrophage heterogeneity is a key feature of the innate immune system even in the absence of adaptive immunity and is conserved more widely across evolution than previously anticipated.

## Materials and methods

N.b. Key Resources Table can be found in Appendix 1 at the end of the manuscript.

### Fly genetics and reagents

Standard cornmeal/agar/molasses media was used to culture *Drosophila* at 25°C (see *Supplementary file 2* for ingredients). *srpHemo-GAL4* (*Brückner et al., 2004*; *Wood et al., 2006*), *crq-GAL4* (*Stramer et al., 2005*), and the *GAL4*-independent lines *srpHemo-GMA* (received from J. Bloor, University of Kent, UK), *srpHemo-3x-mCherry* and *srpHemo-H2A-3x-mCherry* (*Gyoergy et al., 2018*) were used to label the entire hemocyte population during embryonic development or in adults. N.b. *SrpHemo-GAL4* is referred to as *srp-GAL4* on graphs (for reasons of space) but this is the shorter construct more specific to hemocytes (as per *Brückner et al., 2004*) rather than the entire *serpent* promoter region. *Hml(Δ)-GAL4* (*Sinenko and Mathey-Prevot, 2004*) was used to

label larval hemocytes and *da-GAL4* (*Wodarz et al., 1995*) was used as a ubiquitous driver line. These GAL4 lines, Vienna Tiling array GAL4 lines (*VT-GAL4* lines obtained from VDRC; *Kvon et al., 2014*) and split GAL4 lines (see below) were used to drive expression from *UAS-tdTomato*, *UAS-GFP* (*Stramer et al., 2005*; *Wood et al., 2006*), *UAS-red stinger* (*Davis et al., 2012*), *UAS-stinger*, *UAS-Cnx14D*, *UAS-GFP-myc-2xFYVE* (*Wucherpfennig et al., 2003*), or *UAS-GC3ai* (*Schott et al., 2017*). GAL4-independent *VT-RFP* lines were also generated as part of this study (see below) and used to label subpopulation cells in combination with *crq-GAL4,UAS-GFP* (*Stramer et al., 2005*), *eater-GFP* (*Sorrentino et al., 2007*), and *simu-cytGFP* (*Kurant et al., 2008*). Experiments were conducted in a *w$^{1118}$* background and the *repo$^{03702}$* null allele was used to expose plasmatocytes to enhanced levels of apoptotic cell death in the embryo (*Armitage et al., 2020*; *Campbell et al., 1994*; *Halter et al., 1995*). Both *UAS-tdTomato* and *UAS-GFP* were used to analyse subpopulations in the developing embryo in order to ensure labelling of discrete numbers of plasmatocytes was not due to positional effects of insertion sites that led to mosaic expression (*Figure 2*). G-TRACE flies (*w;;UAS-red stinger,UAS-FLP,Ubi-p63E(FRT.STOP)Stinger*; *Evans et al., 2009*) were crossed to split GAL4 driver lines (see below) for lineage-tracing experiments. See *Supplementary file 1* for a full list of *Drosophila* genotypes, transgenes and the sources of the *Drosophila* lines used in this study.

Flies were added to laying cages attached to apple juice agar plates supplemented with yeast paste and allowed to acclimatise for 2 days before embryo collection. Plates were then changed every evening and cages incubated at 22°C overnight before embryos were collected the following morning. Embryos were collected by washing the plates with distilled water and gently disturbing the embryos with a paintbrush, after which embryos were collected into a cell strainer. Embryos were dechorionated in undiluted bleach for 1–2 min and then washed in distilled water until free from bleach. The fluorescent balancers *CTG*, *CyO dfd*, *TTG*, and *TM6b dfd* (*Halfon et al., 2002*; *Le et al., 2006*) were used to discriminate homozygous embryos after removal of the chorion.

## Generation of split GAL4 and GAL4-independent transgenic lines

We used the split GAL4 system (*Pfeiffer et al., 2010*) to restrict VT enhancer expression to *serpent*-positive cells. The activation domain (AD) of *GAL4* was expressed using a well-characterised fragment of the hemocyte-specific *serpent* promoter (*Brückner et al., 2004*; *Gyoergy et al., 2018*) and the DNA-binding domain (DBD) was expressed under the control of VT enhancer regions corresponding to *VT17559-GAL4*, *VT32897-GAL4*, *VT57089-GAL4*, or *VT62766-GAL4*. High-fidelity polymerase (KAPA HiFi Hotstart ReadyMix, Roche) was used to PCR amplify VT enhancer regions from genomic DNA extracted from the original *VT-GAL4* line flies, which were then TA cloned into the *pCR8/GW/TOPO* vector. Primers were designed according to VT enhancer sequences available via the Stark Lab Fly Enhancers website (http://enhancers.starklab.org/; *Kvon et al., 2014*). To make *VT-DBD* transgenic constructs, VT enhancers were transferred from *pCR8/GW/TOPO* into *pBPZpGal4DBDUw* (Addgene plasmid 26233) using LR clonase technology (Gateway LR Clonase II Enzyme Mix, Invitrogen).

To express the *DBD* and *AD* of *GAL4* under the control of the *serpent* promoter (*srpHemo-AD* and *srpHemo-DBD*; also referred to as *srp-AD* and *srp-DBD* for reasons of space on graphs), these were subcloned into a vector containing an *attB* site and this promoter (*pBS_MCS_SRPW_attB*; DSPL337 – a gift from Daria Siekhaus, IST, Austria; *Gyoergy et al., 2018*). *DBD* and *AD* sequences along with the *Drosophila* synthetic minimal core promoter (DSCP) region were amplified using PCR from vectors *pBPZpGal4DBDUw* and *pBPp65ADZpUw* (Addgene clone 26234) using primers that added NotI and AvrII restriction sites (CTGATCGCGGCCGCAAAGTGGTGATAAACGGCCGGC and GATCAGCCTAGGGTGGATCTAAACGAGTTTTTAAGCAAACTCAC). These were subcloned into DSPL337 cut with NotI/AvrII (New England Biolabs) using T4 DNA ligase (Promega). Transgenic flies were generated by site-specific insertion of transgenic constructs into the *VK1 attP* site on chromosome 2 and/or *attP2* on chromosome 3 (Genetivision).

To generate GAL4 independent *VT-RFP* transgenic lines, *nuclear RFP* was isolated by sequential digestion of *pRed H-Pelican* (DGRC plasmid 1203) using Acc65I and then SpeI restriction enzymes (NEB). In parallel, *GAL4* was excised from *pBPGUw* (Addgene plasmid 17575) using the same restriction enzymes and replaced with *nuclear RFP* using T4 ligase. LR clonase was again used to transfer the VT enhancer regions from the *PCR8/GW/TOPO* gateway vectors (see above) into the *nuclear RFP*-containing *pBPGUw* destination vector. Transgenic flies were generated by PhiC31 integrase-mediated insertion of *VT-RFP* constructs into *attP2* on chromosome 3 (Genetivision).

## Imaging of *Drosophila* embryos, larvae, pupae, and adults

Live embryos were mounted ventral-side up on double-sided sticky tape in a minimal volume of Voltalef oil (VWR), after dechorionation in bleach as previously (*Evans et al., 2010*). High-resolution live imaging of plasmatocytes was carried out on an UltraView Spinning Disk system (Perkin Elmer) using a 40x UplanSApo oil immersion objective lens (NA 1.3). A Nikon A1 confocal microscope was used to image plasmatocyte morphology (40x CFI Super Plan Fluor ELWD oil immersion objective lens, NA 0.6) and a Zeiss Airyscan microscope (40x Plan-Apochromat oil immersion objective lens, NA 1.4) was used for imaging of embryos stained with ROS dyes.

L1 and L2 larvae were allowed to develop at 22°C from embryos laid on apple juice agar plates at the same temperature. Larvae were selected and washed in distilled water in embryo baskets, then partially anaesthetised using diethyl ether (2 min for L1 larvae, 3.5 min for L2 larvae). Larvae were then transferred to double-sided tape and covered with halocarbon oil 500 (Sigma-Aldrich). Thickness one coverslip bridges (VWR) were attached to the tape either side of larvae and another coverslip placed across these supports (over the larvae) and attached in place with nail varnish. Larvae were immediately imaged on a MZ205 FA fluorescent dissection microscope with a 2x PLANAPO objective lens (Leica) and LasX software (Leica). To quantify numbers of cells labelled via the split GAL4 system in newly hatched larvae, L1 larvae were flattened under a coverslip in a small drop of halocarbon oil 500. Overlapping images of the flattened larvae were taken using the same microscope and mosaics assembled in Adobe Photoshop. Mosaics were blinded and the number of cells expressing Stinger counted using the multipoint selection tool in Fiji. The same microscope was also used to image L3 larvae, white pre-pupae (WPP), pupae, and adults (see below).

Wandering L3 Larvae and WPP were removed from straight-sided culture bottles containing the food on which they were reared at 25°C and cleaned in distilled water. L3 Larvae were imaged in fresh ice-cold, distilled water to minimise their movements, while WPP were immobilised on double-sided tape (Scotch). For analysis of plasmatocyte populations in pupae, white pre-pupae were also collected, aged at 25°C and the pupal cases removed at a range of times after puparium formation. Dissected pupae were covered with halocarbon oil 500 to prevent desiccation during imaging. For imaging of plasmatocyte populations in adults, females were aged in vials containing cornmeal/agar/molasses media (*Supplementary file 2*) at 25°C, with no more than seven flies kept per vial. Flies were transferred to new food vials every 2–3 days; flies were chilled at −20°C for 4 min and imaged in a petri dish on top of ice to minimise their movements.

## Analysis of hemocyte distribution in larvae

To analyse the distribution of subpopulation cells along the body axis in L3 larvae, the relative proportions of cells within the anterior, medial (abdominal segments A3-A5), and posterior regions were calculated. The number of cells in each region in images of L3 larvae were counted in Fiji and expressed as a fraction of the total number of cells in each larva. As a comparison to reflect the distribution of the total larval hemocyte population, images of L3 larvae with hemocytes labelled using *Hml(Δ)-GAL4,UAS-GFP* were analysed. Since hemocytes were too numerous to count accurately in these images, the integrated density of GFP fluorescence (mean gray value multiplied by area) was measured in each region in Fiji. The proportion of the total GFP signal in each region was then calculated per larva.

## Dissection, stimulation, and staining of larval and adult hemocytes

To isolate larval hemocytes, single wandering third instar were picked from bottles with a paintbrush, washed with distilled water then placed in a 75 μL drop of ice-cold S2 media, which consists of Schneider's media (Sigma-Aldrich) supplemented with 10% heat-inactivated FBS (Gibco/Sigma-Aldrich) and 1x Pen/Strep (Gibco/Sigma-Aldrich). Larvae were then ripped open from the posterior end using size five forceps to release hemocytes into the S2 cell media. Larval carcasses were gently agitated for 5 s before being removed from the S2 media droplet. The cell suspension was then transferred to a well of a 96-well plate (Greiner) and a further 75 μL of S2 media was added per well. Hemocytes were allowed to settle for at least 90 min in a humidified box prior to fixation. Cells were then fixed for 15 min using 4% EM-grade formaldehyde (Thermo Scientific) in PBS (Oxoid). Cells were then permeabilised for 4 min using 0.1% Triton-X-100 in PBS. Following washing in PBS, nuclei and actin filaments were stained using NucBlue (two drops per ml; Invitrogen) and Alexa Fluor 647

phalloidin (1:200 in PBS; Invitrogen; *Figure 7f*) or Alexa Fluor 568 phalloidin (1:500 in PBS; Invitrogen; *Figure 4g*) for 30 min. Following a final wash step, cells were imaged using an ImageXpress Micro hi-content microscope (Molecular Devices).

To discriminate sessile and circulating populations of hemocytes, we adapted a previously described protocol (*Petraki et al., 2015*): wandering third instar larvae were selected, washed and placed in S2 media as above. Larvae were then bled by puncturing at their posterior and anterior ends with sterile 27G needles to release circulating hemocytes; larvae were not agitated and were left in the media droplet for up to 10 s. The bled larva was then transferred to another 75 µL drop of S2 media, while the media containing the initial bleed was transferred to a well in a 96-well plate. Bled larval carcasses were held down using a 27G needle and then jabbed/scraped with a separate needle to release sessile/adherent hemocytes. After jabbing/scraping, the carcass was removed, and the media droplet transferred to a well of a 96-well plate (Greiner). Finally, 75 µL of S2 media was added to each well and cells were allowed to settle prior to being fixed, stained and imaged as above.

For stimulation with *S. cerevisiae* (*Figure 1*), several larvae were pooled and dissected within larger droplets (75 µl per larvae used). Of this cell suspension, 75 µl was transferred into each well in a 96-well plate (Porvair) and cells allowed to adhere in a humidified box in the dark for 2 hr. After 2 hr, cells were stimulated with heat-killed *S. cerevisiae* particles previously stained using calcofluor staining solution (Sigma-Aldrich). *S. cerevisiae* (strain BY4741/accession number Y00000, Euroscarf consortium) were grown to exponential phase in YPD broth (Fisher) at 28°C. Yeast were heat killed at 60°C for 30 min, spun down and frozen at $20 \times 10^9$ cells/ml. $1 \times 10^9$ heat-killed yeast particles in 1 ml of PBS were stained for 30 min at room temperature (with rotation) using 15 µl of calcofluor staining solution. Stained yeast particles were washed in PBS and $1 \times 10^6$ particles resuspended in 75 µl S2 cell medium, which was then added to each well of larval hemocytes for 2 hr. Cells were fixed in wells using 4% EM-grade formaldehyde in PBS for 15 min and washed in PBS. Images were taken on a Nikon Ti-E inverted fluorescence microscope using a 20x objective lens and GFP and DAPI filter sets.

To isolate hemocytes from adults, two flies per genotype (1 day post-eclosion) were anaesthetised using $CO_2$ and cut in half longitudinally in a 75 µl droplet of S2 media on ice. A further 75 µl of S2 media was then added and carcasses agitated by pipetting for 10 s to release hemocytes. The 150 µl of cell suspension was then transferred to a single well in a 96-well plate (Greiner). Cells were allowed to settle for 30 min before being fixed and stained as per larval hemocytes. Based on phalloidin staining and cell morphology it was assumed all adhered cells were blood cells. The percentage of labelled cells was calculated using the number of Stinger-positive cells divided by the total number of cells in images (NucBlue labelling).

## Wounding assay

Live stage 15 embryos were prepared and mounted as described above. The ventral epithelium of the embryos was ablated on the ventral midline using a Micropoint nitrogen-pulsed ablation laser (Andor) fitted to an Ultraview spinning disk confocal system (PerkinElmer) as previously (*Evans et al., 2015*). Pre-wound z-stacks of 30 µm were taken of superficial plasmatocytes with a 1 µm z-spacing between z-slices. Post-wound images were taken on the same settings either at 2 min intervals for 60 min (*Figure 1*) or at the end timepoint of 60 min (*Figure 8* and *Figure 9*).

The proportion of plasmatocytes labelled with *UAS-stinger* (expression via *srpHemo-GAL4* or *VT-GAL4*) was assessed by counting the number of labelled cells at or in contact with the wound site within a 35 µm deep volume on the ventral midline at 60 min post-wounding; this was divided by the total number of labelled cells present within the stack to calculate the percentage of plasmatocytes responding to injury. The brightfield channel was used to visualise the wound margin and only those embryos with wounds between 1000 µm² and 4000 µm² were included in analyses. Quantification was performed on blinded images in Fiji.

## Quantification of migration speeds/random migration

Embryos were prepared and mounted as previously described (*Evans et al., 2010*). Random migration was imaged using an Ultraview spinning disk system (PerkinElmer), with an image taken every 2 min for 1 hr with a z-spacing of 1 µm and approximately 20 µm deep from the ventral nerve cord

using a 20x UplanSApo air objective lens (NA 0.8). Maximum projections were made for each time-point (25 μm depth) and the centre of individual plasmatocyte cell bodies tracked using the manual tracking plugin in Fiji. Random migration speed (μm/min) and directionality (the ratio of the Cartesian distance to the actual distance migrated) were then calculated using the Ibidi chemotaxis plugin.

## Quantification of apoptotic cell clearance

The number of apoptotic cell-containing phagosomes per plasmatocyte (averaged per embryo) was used as a read-out of apoptotic cell clearance as previously described (*Evans et al., 2013*). Phago-cytic vesicles were counted using z-stacks of GFP-labelled plasmatocytes taken from live imaging experiments. Phagosomes were scored in the z-slice in which each macrophage exhibited its maximal cross-sectional area. Only labelled plasmatocytes present on the ventral midline of stage 15 embryos were included. Analysis was performed on blinded image stacks. This analysis does not report the absolute numbers of apoptotic corpses per cell but provides a relative read-out of the phagocytic index.

To assay rates of phagocytosis of apoptotic cells, a phosphatidylinositol-3-phosphate reporter (*UAS-GFP-myc-2xFYVE*; *Roddie et al., 2019*; *Wucherpfennig et al., 2003*) was expressed in all plas-matocytes (via *srpHemo-AD* in combination with *srpHemo-DBD*) or in subpopulation cells (via *srpHemo-AD* in combination with VT enhancers driving expression of the DBD domain). Plasmato-cytes were imaged at stage 12/13 on the ventral midline and the number of 'FYVE events' (number of times new recruitment events could be seen to form on the surface of nascent phagosomes) per plasmatocyte per movie was scored. Only cells present in at least 15 min of 30 min movies were included in this analysis and scoring was conducted on blinded movies constructed from maximum projections of z-stacks.

## Morphological analysis of plasmatocytes

For morphological analysis of plasmatocytes (*Figure 8—figure supplement 1*), the vitelline membrane was manually removed from individual z-slices by drawing around the inside edge of the membrane with the freehand selection tool and using the clear outside command. Maximum projections were then created of the ventral midline region in Fiji. Following this, a region of interest was manually drawn around the area of individual plasmatocytes using the polygon tool and a range of cell shape descriptors and measurements calculated using Fiji.

## ROS staining of embryos

To stain plasmatocyte ROS levels (*Figure 8—figure supplement 2*), embryos containing tdTomato-labelled plasmatocytes (with expression driven using *srpHemo-GAL4* or *VT-GAL4*) were first dechor-ionated and then left in water for 30 min. Stage 15 embryos were then selected and transferred to a glass vial wrapped in foil containing 1 ml peroxide-free heptane (Sigma-Aldrich) and 1 ml of 50 μM dihydrorhodamine 123 (DHR123, Sigma-Aldrich) in PBS. Embryos were shaken at 250 rpm for 30 min. Following this, embryos were removed from the interface and mixed with halocarbon oil. Embryos were orientated individually in a droplet of this oil on a glass slide and then immediately imaged using a Zeiss Airyscan microscope (40x Plan-Apochromat oil immersion objective, NA 1.4), with z-spacing of 1 μm and stacks totalling 30 μm from the surface of the vitelline membrane down through the ventral nerve cord. Embryos were exposed to 10 mM of $H_2O_2$ (Sigma-Aldrich) in PBS and peroxide-free heptane for 30 min prior to staining with DHR123 as a positive control. Negative control embryos were incubated in heptane/PBS alone.

To quantify ROS levels, the intensity of DHR123 staining was measured in the z-slice in which each macrophage exhibited its maximal cross-sectional area. The body of the macrophage was drawn around using the polygon tool in Fiji and then the area and mean gray value were measured in the GFP (DHR123) channel. Average mean gray value per plasmatocyte, per embryo was then plotted in Prism.

## Phagocytosis of *E. coli*

To assay phagocytosis of an immune challenge (*Figure 8—figure supplement 3*), dechorionated stage 15 embryos were mounted ventral-side up on a slide using double-sided Scotch tape, then

dehydrated by incubating in a small container with silica beads for 7–8 min. Further dehydration was then prevented by covering embryos in a small drop of Voltalef oil. 1 mg/ml pHrodo green *E. coli* BioParticles (Invitrogen; resuspended in PBS) were microinjected into the anterior of stage 15 embryos to determine the phagocytic capability of labelled plasmatocytes. Needles were created by pulling 15 cm long 1 mm glass capillaries (World Precision Instruments) using a Flaming/Brown P-1000 micropipette puller (Sutter, program 51). Needle tips were snapped using forceps under high magnification to create a bevelled end. Imaging was performed 1 hr after injection using an UltraView Perkin Elmer Spinning Disk system (40x UplanSApo oil immersion objective lens, NA 1.3). The proportion of *VT-GAL4* or *srpHemo-GAL4*-positive cells containing *E. coli* BioParticles was scored.

## Fixation and immunostaining of embryos

Embryos were fixed and stained as previously described (*Roddie et al., 2019*). Embryos containing plasmatocytes labelled via *srpHemo-GMA* and *GAL4*-driven tdTomato expression were fixed, then mounted in Dabco mountant. Control and *repo* mutant embryos containing plasmatocytes labelled via *crq-GAL4,UAS-GFP* were fixed and immunostained using mouse anti-GFP (ab1218 1:200; Abcam) and rabbit anti-cleaved DCP-1 (9578S 1:1000; Cell Signaling Technologies) to label plasmatocytes and apoptotic cells, respectively. Primary antibodies were detected using Alexa Fluor 488 goat anti-mouse IgG and Alexa Fluor 568 goat anti-rabbit IgG (Invitrogen/Molecular Probes; both used at a 1:400 dilution). Embryos were imaged on the Nikon A1 system described above.

## Image analysis and statistical analysis

All microscopy images were processed using Fiji (*Schindelin et al., 2012*). Images were blinded ahead of analysis with quantification performed on maximum z-projections, with the exception of analysis of numbers of cells labelled via *VT-GAL4* lines (*Figure 2h*), wound responses (*Figure 8c–d*), apoptotic cell clearance (*Figure 8k*) and quantification of ROS staining (*Figure 8—figure supplement 2f*). Quantification was performed on blinded z-stacks for those analyses.

Statistical tests were performed using Prism 7 (GraphPad). p-Values less than 0.05 were deemed significant. Experiments were carried out across at least three independent imaging sessions with N numbers representing individual embryos, with N numbers cited for each condition in the appropriate figure legend. No outliers were excluded. Embryos were taken from laying cages containing greater than 50 adult flies of the parental genotypes, with mutant or control embryos of the correct genotype and developmental stage selected at random following dechorionation. N numbers were typically sufficiently large to enable use of parametric tests. Student's t-test was used when comparing two experimental data sets; where multiple comparisons were required, a one-way ANOVA with Dunnett's multiple comparisons test was performed (parametric data) or the Kruskall-Wallis with Dunn's multiple comparisons test was used (non-parametric data). N numbers, p-values and details of statistical tests employed are reported in the appropriate figure legend. All raw numerical data can be found within the supplementary material as source data files.

## Acknowledgements

This work was funded by a Wellcome/Royal Society Sir Henry Dale Fellowship (102503/Z/13/Z) awarded to IRE and a Bateson Centre PhD studentship awarded to IRE, MPZ and JAC; EB was supported by a PhD studentship from the MRC Discovery Medicine North (DiMeN) Doctoral Training Partnership (MR/N013840/1). Imaging work was performed at the Wolfson Light Microscopy Facility, using the Perkin Elmer spinning disk (MRC grant G0700091 and Wellcome grant 077544/Z/05/Z), Nikon A1 confocal/TIRF (Wellcome grant WT093134AIA) and Zeiss AiryScan microscopes. We thank Steve Brown and the Sheffield RNAi Screening Facility (Wellcome grant 084757) for use of the ImageExpress hi-content microscope and support using this equipment. This work would not be possible without reagents and resources obtained from or maintained by the Bloomington *Drosophila* Stock Centre (NIH P40OD018537), the *Drosophila* Genomics Resource Center (NIH Grant 2P40OD010949), the Vienna *Drosophila* Research Centre and Flybase (NIH and MRC grants U41 HG000739 and MR/N030117/1, respectively). We thank the *Drosophila* community for sharing *Drosophila* reagents (see *Supplementary file 1*). We are grateful to Darren Robinson and Karen Plant and the Fly Facility staff (University of Sheffield) for their support and to Phil Elks and Simon

Johnston (University of Sheffield) for critical reading and feedback on the manuscript. We particularly thank Phil Elks and the now sadly departed Jarema Malicki for their advice and suggestions throughout the project. We are grateful to Rosie Davis and Juliette Howarth for technical assistance with experiments; we thank Adel Alqarni, Agata Grettka, Erin Evans and Eleanor Castle for additional experiments that have contributed to our understanding of this project.

## Additional information

### Funding

| Funder | Grant reference number | Author |
|---|---|---|
| Royal Society | Sir Henry Dale Fellowship 102503/Z/13/Z | Iwan Robert Evans |
| Wellcome | Sir Henry Dale Fellowship 102503/Z/13/Z | Iwan Robert Evans |
| Medical Research Council | MRC Discovery Medicine North (DiMeN) Doctoral Training Partnership MR/N013840/1 | Elliot Brooks |

The funders had no role in study design, data collection and interpretation, or the decision to submit the work for publication.

### Author contributions

Jonathon Alexis Coates, generated transgenic flies, performed experiments, analysed data, contributed to experimental design, wrote the initial manuscript draft and helped revise the manuscript; Elliot Brooks, performed experiments, analysed data, contributed to experimental design and helped revise the manuscript; Amy Louise Brittle, Emma Louise Armitage, generated transgenic flies, performed experiments, analysed data, contributed to experimental design and helped revise the manuscript; Martin Peter Zeidler, conceived the projected, obtained funding, supervised the project, performed experiments, analysed data, contributed to experimental design and helped revise the manuscript; Iwan Robert Evans, conceived the project, obtained funding for the project, supervised the project, performed experiments, analysed data, designed experiments, wrote the initial manuscript draft and revised the manuscript.

### Author ORCIDs

Jonathon Alexis Coates (iD) https://orcid.org/0000-0001-9039-9219
Elliot Brooks (iD) https://orcid.org/0000-0003-1583-0712
Amy Louise Brittle (iD) https://orcid.org/0000-0002-3911-0181
Emma Louise Armitage (iD) https://orcid.org/0000-0003-0880-1948
Iwan Robert Evans (iD) https://orcid.org/0000-0002-3485-4456

### Decision letter and Author response

Decision letter https://doi.org/10.7554/eLife.58686.sa1
Author response https://doi.org/10.7554/eLife.58686.sa2

## Additional files

### Supplementary files

• Supplementary file 1. Table showing genotypes and sources of fly lines used in this study.

• Supplementary file 2. Table showing the fly food recipe used in this study.

• Transparent reporting form

## Data availability

All data generated or analysed during this study are included in the manuscript and supporting files. Source data files have been provided for all raw numerical data.

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

# Appendix 1

**Appendix 1—key resources table**

| Reagent type (species) or resource | Designation | Source or reference | Identifiers | Additional information |
|---|---|---|---|---|
| Gene (*Drosophila melanogaster*) | w | NA | FLYB:FBgn0003996 | NA |
| Gene (*D. melanogaster*) | srp | NA | FLYB:FBgn0003507 | NA |
| Gene (*D. melanogaster*) | crq | NA | FLYB:FBgn0015924 | NA |
| Gene (*D. melanogaster*) | simu | NA | FLYB:FBgn0260011 | Also known as *NimC4* |
| Gene (*D. melanogaster*) | eater | NA | FLYB:FBgn0243514 | NA |
| Gene (*D. melanogaster*) | Cnx14D | NA | FLYB:FBgn0264077 | NA |
| Gene (*D. melanogaster*) | repo | NA | FLYB:FBgn0011701 | NA |
| Genetic reagent (*D. melanogaster*) | $w^{1118}$ | Evans lab stock | FLYB:FBal0018186 | FlyBase symbol:$w^{1118}$ |
| Genetic reagent (*D. melanogaster*) | srpHemo-GAL4 | PMID:15239955 | FLYB:FBtp0023390 | FlyBase symbol:P{srp.Hemo-GAL4}2; Obtained from W. Wood, University of Edinburgh, UK |
| Genetic reagent (*D. melanogaster*) | srpHemo-Gal4, UAS-GFP | PMID:16651377 | FLYB:FBtp0023390 (P{srp.Hemo-GAL4}2) | Obtained from W. Wood, University of Edinburgh, UK |
| Genetic reagent (*D. melanogaster*) | srpHemo-GAL4, UAS-red stinger | PMID:23172914 | FLYB:FBtp0023390 (P{srp.Hemo-GAL4}2) | Obtained from B. Stramer, Kings College London, UK |
| Genetic reagent (*D. melanogaster*) | crq-GAL4,UAS-GFP | PMID:15699212 | FLYB:FBtp0022491 (P{crq-GAL4.A}) | Obtained from W. Wood, University of Edinburgh, UK |
| Genetic reagent (*D. melanogaster*) | da-GAL4 | PMID: FBrf0082789 | FLYB:FBtp0019571 | FlyBase symbol:P{da-GAL4.w-}; Obtained from A. Whitworth, University of Cambridge, UK |
| Genetic reagent (*D. melanogaster*) | UAS-GC3ai | PMID:28870988 | FLYB:FBtp0137390 | FlyBase symbol:P{UAS-GC3Ai} 3; Obtained from M. Suzanne, CBI-Toulouse, France |
| Genetic reagent (*D. melanogaster*) | srpHemo-3x-mCherry | Bloomington *Drosophila* Stock Center; PMID:29321168 | BDSC: 78359; FLYB: FBtp0127793; RRID:BDSC_78359 | FlyBase symbol:P{srpHemo-3XmCherry}; Obtained from B, Stramer, Kings College London, UK |
| Genetic reagent (*D. melanogaster*) | Hml(Δ)-GAL4,UAS-GFP | Bloomington *Drosophila* Stock Center; PMID:15480416 | BDSC:30140; RRID:BDSC_30140; FLYB:FBtp0040877; BDSC:30142; RRID:BDSC_30142; FLYB:FBtp0040877 | FlyBase symbol:'$w^{1118}$; P{Hml-GAL4.Δ}2, P{UAS-2xEGFP}AH2';'$w^{1118}$; P{Hml-GAL4.Δ}3, P{UAS-2xEGFP}AH3/MKRS' |
| Genetic reagent (*D. melanogaster*) | VT17559-GAL4 | PMID:24896182 | VDRC:205658 | Previously available from Vienna *Drosophila* Research Center (stock discarded); available on request from I. Evans |

*Continued on next page*

*Appendix 1—key resources table continued*

| Reagent type (species) or resource | Designation | Source or reference | Identifiers | Additional information |
|---|---|---|---|---|
| Genetic reagent (*D. melanogaster*) | VT32897-GAL4 | PMID:24896182 | VDRC:214064 | Previously available from Vienna *Drosophila* Research Center (stock discarded); available on request from I. Evans |
| Genetic reagent (*D. melanogaster*) | VT57089-GAL4 | PMID:24896182 | VDRC:208119 | Previously available from Vienna *Drosophila* Research Center (stock discarded); available on request from I. Evans |
| Genetic reagent (*D. melanogaster*) | VT62766-GAL4 | PMID:24896182 | VDRC:203897 | Previously available from Vienna *Drosophila* Research Center (stock discarded); available on request from I. Evans |
| Genetic reagent (*D. melanogaster*) | UAS-tdTomato | Bloomington *Drosophila* Stock Center | BDSC:36327; FLYB: FBti0145103; RRID:BDSC_36327 | FlyBase symbol:P{UAS-tdTom.S}2 |
| Genetic reagent (*D. melanogaster*) | srpHemo-GMA | Other | NA | Globular Moesin actin-binding domain fused to GFP under the control of srpHemo; P-element insertions on chromosomes 2 and 3; Obtained from James Bloor, University of Kent, UK |
| Genetic reagent (*D. melanogaster*) | UAS-GFP | Bloomington *Drosophila* Stock Center | BDSC: 5431; FLYB: FBti0013988; RRID:BDSC_5431 | FlyBase symbol:P{UAS-EGFP}5a.2 |
| Genetic reagent (*D. melanogaster*) | eater-GFP | PMID:17936744 | FLYB: FBtp0054463 | FlyBase symbol:P{eater-GFP.1.7}; Obtained from L. Vesala, University of Tampere, Finland |
| Genetic reagent (*D. melanogaster*) | simu-cytGFP | PMID:18455990 | NA | FlyBase symbol:M{simu-cytGFP}; Obtained from E. Kurant, University of Haifa, Israel |
| Genetic reagent (*D. melanogaster*) | VT17559-RFP | This paper | NA | Inserted in *attP2* on chromosome 3; see methods for details of cloning and transgenesis |
| Genetic reagent (*D. melanogaster*) | VT32897-RFP | This paper | NA | Inserted in *attP2* on chromosome 3; see methods for details of cloning and transgenesis |
| Genetic reagent (*D. melanogaster*) | VT57089-RFP | This paper | NA | Inserted in *attP2* on chromosome 3; see methods for details of cloning and transgenesis |
| Genetic reagent (*D. melanogaster*) | VT62766-RFP | This paper | NA | Inserted in *attP2* on chromosome 3; see methods for details of cloning and transgenesis |
| Genetic reagent (*D. melanogaster*) | srpHemo-AD | This paper | NA | Inserted in *VK1 attP site* on; see methods for details of cloning and transgenesis chromosome 3 |

*Continued on next page*

*Appendix 1—key resources table continued*

| Reagent type (species) or resource | Designation | Source or reference | Identifiers | Additional information |
|---|---|---|---|---|
| Genetic reagent (*D. melanogaster*) | srpHemo-DBD | This paper | NA | Inserted in *attP2* on chromosome 3; see methods for details of cloning and transgenesis |
| Genetic reagent (*D. melanogaster*) | VT17559-DBD | This paper | NA | Inserted in *attP2* on chromosome 3; see methods for details of cloning and transgenesis |
| Genetic reagent (*D. melanogaster*) | VT32897-DBD | This paper | NA | Inserted in *attP2* on chromosome 3; see methods for details of cloning and transgenesis |
| Genetic reagent (*D. melanogaster*) | VT57089-DBD | This paper | NA | Inserted in *attP2* on chromosome 3; see methods for details of cloning and transgenesis |
| Genetic reagent (*D. melanogaster*) | VT62766-DBD | This paper | NA | Inserted in *attP2* on chromosome 3; see methods for details of cloning and transgenesis |
| Genetic reagent (*D. melanogaster*) | srpHemo-H2A-3x-mCHerry | Bloomington *Drosophila* Stock Center; PMID:29321168 | BDSC: 78361; FLYB: FBtp0127794; RRID:BDSC_78661 | FlyBase symbol:P{srpHemo-H2A.3XmCherry}; Obtained from B. Stramer, Kings College London, UK |
| Genetic reagent (*D. melanogaster*) | UAS-stinger | Bloomington *Drosophila* Stock Center | BDSC:84277; FLYB: FBti0074589; RRID:BDSC_84277 | FlyBase symbol:P{UAS-Stinger}2 |
| Genetic reagent (*D. melanogaster*) | 'w;;UAS-red stinger,UAS-FLP, Ubi-p63E(FRT. STOP) Stinger' | Bloomington *Drosophila* Stock Center; PMID:19633663 | BDSC:28281; RRID:BDSC_28281 | FlyBase symbol:'w[*]; P{w[+mC]=UAS-RedStinger}6, P{w[+mC]=UAS-FLP.Exel}3, P{w[+mC]=Ubi-p63E(FRT.STOP)Stinger}15F2'; Obtained from Alisson Gontijo, CEDOC, Lisbon, Portugal |
| Genetic reagent (*D. melanogaster*) | UAS-tdTomato | Bloomington *Drosophila* Stock Center | BDSC:36327; FLYB: FBti0145103; RRID:BDSC_36327 | FlyBase symbol:P{UAS-tdTom. S}2 |
| Genetic reagent (*D. melanogaster*) | UAS-GFP-myc-2xFYVE | Bloomington *Drosophila* Stock Center | BDSC:42712; FLYB: FBti0147756; RRID:BDSC_42712 | FlyBase symbol:P{UAS-GFP-myc-2xFYVE}2 |
| Genetic reagent (*D. melanogaster*) | UAS-Cnx14D | Harvard *Drosophila* Stock Center | FLYB:FBal0228355 | FlyBase symbol:P{XP} para$^{d04188}$; Previously available from Harvard *Drosophila* Stock Centre (now discarded); available on request from I. Evans |
| Genetic reagent (*D. melanogaster*) | repo$^{03702}$ | Bloomington *Drosophila* Stock Center; PMID:32796812 | BDSC:11604; FLYB: FBti0003552; RRID:BDSC_11604 | FlyBase symbol:ry$^{506}$ P{PZ}repo$^{03702}$/TM3, ry$^{RK}$ Sb$^1$ Ser$^1$; ry$^{506}$ allele recombined off original stock line |
| Biological sample (*D. melanogaster*) | Embryos, L1-L3 larvae, white pre-pupae, pupae, adults, hemolymph (larval and adult) | NA | NA | NA |

*Continued on next page*

*Appendix 1—key resources table continued*

| Reagent type (species) or resource | Designation | Source or reference | Identifiers | Additional information |
|---|---|---|---|---|
| Biological sample (*S. cerevisiae*) | Isogenic *S. cerevisiae* wild-type yeast strain BY4741 | Euroscarf consortium | Euroscarf: Y00000 | Heat-killed and stained with calcofluor staining solution for use in phagocytosis assay; $1 \times 10^6$ particles added per well in 96-well plate |
| Antibody | Anti-cleaved DCP-1 (Asp216) (Rabbit polyclonal) | Cell Signaling Technologies | Cat# 9578S; RRID:AB_2721060 | IF(1:1000); Primary antibody used to detect apoptotic cells |
| Antibody | Anti-GFP (Mouse monoclonal) | Abcam | Cat# ab1218; RRID:AB_298911 | IF(1:200); Primary antibody used to detect cells expressing GFP |
| Antibody | Alexa Fluor 488 Goat anti-Mouse IgG (Goat polyclonal) | Invitrogen/ Molecular Probes | Cat# A11029; RRID:AB_138404 | IF(1:400); secondary antibody used to detect anti-GFP primary antibody |
| Antibody | Alexa Fluor 568 Goat anti-Rabbit IgG (Goat polyclonal) | Invitrogen/ Molecular Probes | Cat# A11036; RRID:AB_10563566 | IF(1:400); secondary antibody used to detect anti-cleaved DCP-1 primary antibody |
| Software, algorithm | Fiji | PMID:22743772 | RRID:SCR_002285 | |
| Software, algorithm | GraphPad Prism 7 | Graphpad | RRID:SCR_002798 | |
| Other | NucBlue | Invitrogen/ Molecular Probes | Cat# R37605 | NucBlue Live ReadyProbes Reagent (Hoechst 33342); two drops per ml; nuclear stain |
| Other | Calcofluor staining solution | Sigma-Aldrich | Cat# 18909–100 ML-F | 40 µl used to stain $1 \times 10^9$ heat-killed yeast particles in 1 mL PBS |
| Other | Dihydrorhodamine 123 ROS dye | Sigma-Aldrich | Cat# D1054-10MG | ROS dye; used at (50 µM) in PBS to stain embryos |
| Other | pHrodo green *E. coli* BioParticles | Invitrogen/ Molecular Probes | Cat# P35366 | Microinjected into embryos at (1 mg/ml) |
| Other | Alexa Fluor 647 phalloidin | Invitrogen/ Molecular Probes | Cat# A22287; RRID:AB_2620155 | 1:200 Dilution |
| Other | Alexa Fluor 568 phalloidin | Invitrogen/ Molecular Probes | Cat# A12379 | 1:500 Dilution |

