## [Decision Letter]

**Acceptance summary:**

*Drosophila* plasmatocytes, which are most akin to macrophages in vertebrates, have been considered as a single population until very recently. This study identified molecularly and functionally distinct subpopulations of plasmatocytes and their plasticity across development. This work will highly contribute to understanding the nature of hemocyte diversity.

**Decision letter after peer review:**

Thank you for submitting your article "Identification of functionally-distinct macrophage subpopulations regulated by efferocytosis in *Drosophila*" for consideration by *eLife*. Your article has been reviewed by 3 peer reviewers, one of whom is a member of our Board of Reviewing Editors, and the evaluation has been overseen by Utpal Banerjee as the Senior Editor. The reviewers have opted to remain anonymous.

The reviewers have discussed the reviews with one another and the Reviewing Editor has drafted this decision to help you prepare a revised submission.

Summary:

In this manuscript, Coates et al. identify distinct subtypes of hemocytes by the Vienna tile enhancer-trap screen and propose the functional divergence of hemocyte subpopulations in the *Drosophila* model organism. The authors label hemocyte with enhancer trap Gal4 drivers and identify Gal4 drivers that exhibit differential expressions in a subset of hemocytes. The Gal4-expressing subpopulations show dynamic patterns during embryonic- and larval development and in adult flies. The subpopulations are further characterized by testing their engagement in wound response, phagocytosis, or migratory activities. The authors also observe that enhancer trap expression in subpopulations changes under conditions such as enhanced organismal apoptosis. Finally, the authors show that cnx14D as an associated gene that promotes wound response in one of the subpopulations found.

This study utilizes an interesting genetic approach to understanding the heterogeneity of hemocytes and suggests the functionally distinct hemocyte subtypes in different stages of *Drosophila*. Despite the potential merits of this study, the reviewers agreed that more wide-reaching and solid conclusions could be drawn by re-analyses of data and thorough and thoughtful rewriting. Many of the essential revisions can be adequately dealt with by careful rewriting and reanalyses and there are additional experiments required to sufficiently address the concerns. I have attached the reviewers' original comments below for details.

Essential revisions:

1. The loss of certain subpopulations shown at larval stages could be due to the disappearance of the subpopulation, but it could be also caused by changes in the enhancer activity. This point is not clear in the manuscript and requires better interpretations. As indicated in the reviewers' concerns (Reviewer 1 concern 1-2, Reviewer 2 concern 2), additional analyses with new experiments (such as GTRACE or Flp out) will clarify the issue.

2. Although the authors identified the subpopulations according to the differential patterns of VT-gal4, the molecular and biological characteristics of these cells need to be better established (Reviewer 1 concern 1-1 and Reviewer 3). Additional experiments will be essential for addressing the point by staining with representative antibodies (e.g. anti-Crq, anti-SIMU, anti-Draper) or enhancers.

3. As suggested by Reviewer 1 concern 3 and Reviewer 2 concerns 5-6, some of the expression patterns indicated by VT-gal4 in larvae or adults require additional analyses and quantitations.

4. Related to point 3, the characterization of larval hemocyte subpopulations need to be substantiated as indicated in Reviewer 1, concern 3 and Reviewer 2, concern 7. Since larval hemocytes relocate between local microenvironments and circulation, looking into earlier larval stages when hemocytes are more resident or proper bleeding of larval hemocytes will resolve the issue.

5. Related to point 3, detailed expression patterns of adult hemocytes can be better presented by providing closeup images (please see Reviewer 2, concern 5-6).

6. Given that hemocytes dynamically relocate and interact with other tissues, it will be critical to address as to how subpopulations shape during development and how they possibly interact with local microenvironments, as suggested in Reviewer 2, concern 1. One way of resolving this concern is to add discussions about recent relevant studies, as indicated in the reviewers' comments. And if possible, related experiments can be performed to sufficiently address the concern.

7. Related to point 6, recent scRNA-seq studies proposed subtypes/subpopulations of hemocytes based on differential transcriptome profiles. The authors can suggest and correlate the identified subtypes with previous analyses in the discussion (as suggested by Reviewer 2, concern 1).

8. As pointed in Reviewer 2 concern 4, srp-gal4 is relatively weak in larval hemocytes and may not be the most appropriate enhancer to use in the stage. It seems it is too late to repeat every experiment with an alternative driver. However, I would suggest to sufficiently discuss the issue in the discussion.

9. This is an optional point. Two of the reviewers suggested reconsidering the title.

Reviewer #1:

Evans and colleagues used VT enhancer reporter lines to label plasmatocyte subpopulations and observed distinct subpopulations that are varied over development. Then the authors take on to characterize differential functions of these cells and found that each subpopulation exhibits variable wound responses, migratory activities, and reactivity to cell death. Further, the authors showed that overexpression of cnx14D, which is coupled with one of the VT-gal4, improves wound responses similar to the cnx14D-associated enhancer.

This timely study suggests that hemocytes are more heterogeneous than has been thought. Moreover, functional analyses highlight that each subpopulation may exhibit differential activities to immune/wound responses. Related to recent single-cell studies where different subtypes of hemocytes are shown, this study will provide functional insights into how each plasmatocyte subpopulation behaves differently under diverse responses. Overall, I found this study interesting and will be of broad interest to the immunity/hematopoiesis field, however, there are several points that need to be substantiated with regard to the characterization of subpopulations and functional analyses of cnx14D.

1. Characterization of the subpopulation

a. In the recent scRNA seq analyses, the subpopulation/subtype/subclass of hemocytes was defined by differential expressions of multiple genes or by transcriptional activities. In the other study, subtypes of lamellocytes derived upon wasp infestation were validated by multiple reporter lines as well as protein expressions (Anderl et al., 2016). Is the expression of VT enhancer alone sufficient for defining the subpopulation? Co-expression of VT+ subpopulations with other markers and quantitation of each population would clarify and strengthen the main findings of this work.

b. The authors showed that VT+ subpopulations change their numbers during the embryonic development, and most of the subpopulations are significantly reduced or even absent in the 3rd instar larvae. These patterns can be explained by changes in the enhancer activity and may not reflect the loss of specific cells. Are hemocytes that had turned on a VT-gal4 remain afterward (larval to WPP) while shutting off its expression? Or do they disappear or die? This question can be in part addressed by the tracing experiment (UAS-Gtrace).

c. We can predict possible target genes of each VT enhancer by their genomic locations. However, enhancer activity does not always reflect the expressions of the associated gene. The authors showed the functional correlation of VT62766 cells with cnx14D without providing evidence for cnx14D expression in the VT62766 subpopulation. Is cnx14D found in the recent scRNA seq data? If so, how does it correlate to (number, developmental stage) the pattern shown by VT62766? If not, can it be validated by mRNA FISH?

2. The authors showed that overexpression of cnx14D in most hemocytes enhances wound responses with more plasmatocytes responding. This phenotype could be due to altered Ca^2+^ dynamics given the function of Ca^2+^ in the wound response as the authors mentioned and does not sufficiently prove that enhanced wound response seen in VT62766+ cells is caused by cnx14D. Does cnx14D RNAi reduce the wound response of VT62766+ hemocytes? And how do the authors expect the 3' end enhancer function in the cnx14D expression or function?

3. Hemocyte populations in larvae and adults need better quantitation and characterization (Figure 3-5). As the whole larvae image only displays the sessile portion of hemocytes, it should be accompanied by proper bleeding of hemocytes. Bleeding hemocytes before and after vortexing, in case of larvae, will separate sessile and circulating hemocyte pools and quantitation of each VT+ cells per total DAPI or other hemocyte markers will provide a clear overview of each subpopulation. Similarly, adult hemocytes can be quantified after bleeding. Moreover, hemocytes attached to each body part can be quantified as shown in Bosch et al. 2019. Since the authors claim that each subpopulation is closely associated with specific sites, this should be quantified.

Reviewer #2:

In the model organism *Drosophila*, Coates et al. identify functionally distinct macrophage (plasmatocyte) populations. Findings will inform *Drosophila* hematopoiesis, immunity, and wound healing. The authors label macrophage subpopulations through enhancer element GAL4 drivers, which show dynamic patterns of expression during development and in the ageing adult fly. Testing by criteria such as wound responses, and phagocytosis of apoptotic cells, the authors find differential engagement of certain macrophage subpopulations. They further find that reporter expression in subpopulations changes during development and shifts under conditions such as enhanced organismal apoptosis. Lastly, the authors provide proof of principle how this enhancer collection can lead to gene discovery: they identify calnexin 14D as a gene that promotes wound responses.

Overall this study is based on an interesting approach. Experiments are carefully conducted. However, in the current state the study misses more wide-reaching conclusions. In some cases the authors seem to make rather unlikely speculations, which should be tested in order to draw solid conclusions. Also, existing literature should be taken more into consideration and cited accordingly.

1. While the authors have surveyed the enhancer-based lines quite broadly, the manuscript does not convey a clear take-home message. What do we really learn from this study functionally? For example, are hemocyte populations shaped by their environment and function, or vice versa do they home to their anatomical sites driven by their expression makeup? While the authors seem to gravitate toward the latter explanation, I think the former is much more likely, also because there is precedent in vertebrates: Lavin et al. Cell 2014, and Gosselin et al. Cell 2014 showed that tissue-resident macrophage enhancer landscapes, and tissue-specific macrophage identities, are shaped by local microenvironments. I think it would be important to investigate this question in more detail, so the authors can draw conclusions based on their findings, whichever way it may be.

2. During development and in the adult, the authors find that certain reporters cease to be detected; likely these hemocyte subpopulations are not lost but the reporter expression is simply downregulated. However, this needs to be properly analyzed by some kind of lineage tracing, such as GTRACE or flipout-lacZ lineage tracing.

3. Can the authors, based on their findings of functionally distinct hemocyte behaviors, suggest correlation of certain subgroups of this study with differentially expressing clusters in scRNAseq studies that were recently published? How about plasmatocyte clusters that may resemble vertebrate M1 or M2 macrophages?

4. Srp-GAL4 is not the best driver for hemocytes in the larva, and may be weak in many plasmatocytes by itself. With this approach the authors may overlook many cells that form plasmatocyte subpopulations. I think it would make more sense to perform split GAL4 expression with one part of GAL4 expressed under Hml∆ control.

5. In the adult, do the expression domains really overlap with the general srp expression? It doesn't look like that to me in some of the current images, and it would be helpful if this could be shown in closeups, and highlighted in the images. Also, some kind of model that would indicate the different populations would be helpful.

6. The authors restrict quantification of hemocyte subsets to the population of hemocytes in the VNC. Would the relative fraction of hemocytes be different elsewhere?

7. Late 3rd instar larvae were used to release hemocytes. However since at that point hemocytes already enter circulation, their defined expression properties may start to be lost, if indeed hemocyte subgroups are defined by factors from local microenvironments. Perhaps it would be beneficial to look earlier (second or early third instar), when hemocytes are still more resident?

Reviewer #3:

This is a solid study, which reveals that *Drosophila* plasmatocytes (macrophages) are not a uniform population as it was considered previously. The authors identified and characterized distinct subpopulations of the fly macrophages, which exhibit molecular and functional differences, and specific localization during different developmental stages. This exciting finding establishes heterogeneity in the innate immune system, which is not connected to adaptive immunity. The authors also reveal that subpopulation sizes are affected by increased apoptosis, demonstrating plasticity of the system. In addition, genes associated with the subpopulation-specific enhancers might act as the effector genes of the specific function, which is nicely exemplified with overexpression of calnexin14-D and opens new directions in clarifying the mechanisms regulating macrophage diversity.

I am reluctant to ask for additional experiments but it would be interesting to test whether other known genes involved in efferocytosis by macrophages like crq show differential expression in distinct subpopulations. It could be examined by immunostaining of different VT-Gal4>GFP labeled populations with anti-Crq, anti-Draper and anti-SIMU antibodies.

In my view, the paper is suitable for publication in *eLife* as it stands.

---

## [Author Response]

Essential revisions:1. The loss of certain subpopulations shown at larval stages could be due to the disappearance of the subpopulation, but it could be also caused by changes in the enhancer activity. This point is not clear in the manuscript and requires better interpretations. As indicated in the reviewers' concerns (Reviewer 1 concern 1-2, Reviewer 2 concern 2), additional analyses with new experiments (such as GTRACE or Flp out) will clarify the issue.

In our experience we have not observed apoptosis of hemocytes during larval development. Consistent with this, there is no evidence in the literature that suggests there is significant death of blood cells during larval development that could explain the decrease in numbers of subpopulation cells by the L3 stage. This suggests that subpopulations of labelled cells may be lost via reprogramming to alternative states, which would very likely involve changes in gene expression (that are visualised by our reporters). We therefore interpret the loss of enhancer activity as a loss of subpopulation identity and an example of cellular plasticity, as witnessed in reprogramming of macrophages during inflammatory responses in vertebrates. We have made this important point clearer in the manuscript at various points. In addition, we also present additional lineage tracing experiments we have undertaken that also support this conclusion (see below).

As requested, we have added further characterisation of subpopulations at a number of developmental stages and expanded our characterisation to include larval stages L1 and L2 in a completely new figure (Figure 3). This new data includes quantification of the proportions of blood cells that are positive for VT enhancer expression in L1 larvae (Figure 3c), L3 larvae (new data, Figure 4g-h) and in adults (new data, Figure 7f). This reveals additional information about the dynamics of these subpopulations and identified a dramatic decrease in proportions of labelled cells between L2 and wandering L3 stages of development. The L1 and L2 data are described from page 9, line 198. The L3 data are described from page 10, line 228. The adult data are described and discussed from page 13, line 294.

Furthermore, we have used lineage tracing (G-TRACE) in combination with our split GAL4 lines to investigate reprogramming of subpopulation cells during larval stages in a new figure (Figure 4—figure supplement 1a-b). These data show that VT reporter expression is dynamic and many cells seem to switch off expression during larval development. This supports the hypothesis that these subpopulations are plastic and change across the lifecourse. These data are described from page 11, line 237.

2. Although the authors identified the subpopulations according to the differential patterns of VT-gal4, the molecular and biological characteristics of these cells need to be better established (Reviewer 1 concern 1-1 and Reviewer 3). Additional experiments will be essential for addressing the point by staining with representative antibodies (e.g. anti-Crq, anti-SIMU, anti-Draper) or enhancers.

We have addressed this point through a series of new experiments, visualising subpopulation cells using novel GAL4-independent lines we have made (*VT-RFP*) in combination with enhancer-based reporters of known markers (Crq, Simu, Eater). *eater-GFP* does not overlap with subpopulation cells in the embryo (new figure, Figure 2—figure supplement 1). *simu-cytGFP* expression is uniformly expressed in embryonic plasmatocytes and therefore does not specifically define subpopulation cells (new figure, Figure 2—figure supplement 2a,c). In contrast to *simu* and as shown in Figure 1b-b’, *crq-GAL4* expression is very heterogeneous (new figure, Figure 2—figure supplement 2). Despite this heterogeneity, *crq* expression does not define subpopulation cells by either its presence or absence (Figure 2—figure supplement 1b-c) with cells expressing RFP alone, GFP alone and both fluorophores present for each *VT-RFP* line tested. Taken together this data suggests that VT-labelled subpopulations do not show obvious overlap with the three well-characterised markers we have investigated. This new data is described from page 8, line 162.

3. As suggested by Reviewer 1 concern 3 and Reviewer 2 concerns 5-6, some of the expression patterns indicated by VT-gal4 in larvae or adults require additional analyses and quantitations.

Reviewer 2 asked about numbers of subpopulation cells in regions away from the VNC (point 6). We have now visualised and quantified elsewhere in the embryo using the split GAL4 lines (new data, Figure 3—figure supplement 1b-g). The relative amounts of each subpopulation seem similar in different regions of the embryo (although numbers of cells detected increases as development proceeds since it is difficult to detect subpopulations at earlier stages when expression first begins). This new data is described and discussed from page 8, line 179. Quantification of subpopulations in L1 larvae (new data, Figure 3c), L3 larvae (new data, Figure 4g) and the adult (new data, Figure 7f) are now also provided.

Reviewer 2 also queried overlap between *srp* domains and subpopulations in the adult. There is biological variation between flies such that the position of hemocytes is not completely stereotyped. As a result it is not expected that the positions of cells in control and experimental images will align perfectly. However, we have replaced the original control panel with a better example in which the thoracic cells are more in focus and have used alternative views taken of the same fly to highlight leg and proboscis hemocytes (Figure 7a). Close ups are also now provided (Figure 7a-e).

Reviewer 1 asked us to investigate *Cnx14D* expression: at least three very closely related genes encode Calnexin proteins in the *Drosophila* genome: *Cnx14D*, *Cnx99A* and *CG1924*, complicating probe design for in situ hybridisation approaches. Nonetheless, we generated a collection of fluorescently-labelled RNA oligonucleotide probes using Stellaris probe design software that were specific to *Cnx14D* which should target neither *Cnx99A* nor *CG1924*. A probe library for *EGFP* was also generated as a positive control. While we were able to detect *EGFP* expression within embryos containing macrophage-specific expression of *EGFP* (data not shown), *Cnx14D* signals were considerably weaker and barely detectable above background / noise (data not shown). This suggests either that our probes did not work or that *Cnx14D* transcripts are expressed at levels below the detection threshold. Interestingly, genome-wide transcriptional data generated by the modENCODE consortium (Graveley et al., 2011 Nature) reports low transcript levels for this gene within embryos consistent with our difficulties detecting *Cnx14D* expression via in situ hybridisation. Sadly, the only two published enhancer trap reporters we were able to identify (*Cnx14D^PG156^* and *Cnx14D^PL42^*; Bourbon et al., 2002 Mech Dev) are no longer extant, having been discarded in spring 2020 (Alain Vincent, personal communication).

We also attempted to address this at the protein level via immunostaining. There is no *Drosophila*-specific antibody available and the anti-hCalnexin antibody that we were able to source (Rb anti-Calnexin ab75801, Abcam) did not appear to detect the fly protein (data not shown). As such, we have tried multiple means to profile expression at the RNA and protein level, but unfortunately to make progress will require further new reagents or a different approach that is beyond the scope of these revisions.

Finally, we interrogated single-cell sequencing data (only openly available for two studies: Tattikota et al., 2020 and Cattenoz et al., 2020). Both studies applied scRNAseq to cells bled from L3 larvae, the stage at which subpopulation cells are least apparent. Cattenoz et al. did not find *Cnx14D* transcripts, although expression within one cluster (PM6) was identified by Tattikota et al. However, this expression was not robust with only 2% of cells in this cluster expressing *Cnx14D* (11/500 cells). Therefore, it is possible that *Cnx14D* produces low abundance transcripts which make detection difficult via both in situ hybridization and scRNAseq.

4. Related to point 3, the characterization of larval hemocyte subpopulations need to be substantiated as indicated in Reviewer 1, concern 3 and Reviewer 2, concern 7. Since larval hemocytes relocate between local microenvironments and circulation, looking into earlier larval stages when hemocytes are more resident or proper bleeding of larval hemocytes will resolve the issue.

We have imaged subpopulations in L1 and L2 larvae and quantified numbers in the former (new Figure 3). We have also made videos of L1 larvae (new Video 2, cited page 9, line 208). While substantial numbers of blood cells are labelled in L1 larvae, fewer are labelled than in *serpent* controls (new Figure 3a-c). Live videos of L1 larvae (new Video 2) show that the majority of VT-labelled cells are attached to the body wall and therefore do not change their relative positions as the larvae contract and attempt to crawl. The large numbers of subpopulation cells (roughly 50% of the total population) in L1 and L2 larvae makes localisation beyond what we have described difficult to discern. This data is described from page 9, line 198.

We have also made additional videos to cover each genotype at L3 stages and tracked cells to show that at least some subpopulation cells are in circulation (new Video 4), with many of the subpopulation cells appearing to be attached to the body wall. This data is described from page 10, line 224.

We have performed bleeds as suggested and characterised relative numbers of sessile and circulating cells and this new data is presented in Figure 4g. This substantiates the imaging showing subpopulation cells are present both in circulation and attached to the body wall, since similar proportions are present in initial bleeds and following scraping of larval carcasses to retrieve sessile cells (as per the method of Petraki et al., 2015 JOVE). This data is described from page 10, line 228.

The numbers of larvae showing the localisations described in the L3 stages are now shown in Figure 4c-f as annotations within the respective figure panels.

Lastly, we have quantified the localisation of the subpopulation cells along the body axis and this data suggests there is a bias in the localisation towards the posterior end compared to localisations of the overall population (Figure 4h-h’). This is described from page 10, line 233.

5. Related to point 3, detailed expression patterns of adult hemocytes can be better presented by providing closeup images (please see Reviewer 2, concern 5-6).

We now provide close ups of the key images and extra annotation (Figure 7a-e). See Point 3 above for our response regarding overlap with *srp* expression domains.

6. Given that hemocytes dynamically relocate and interact with other tissues, it will be critical to address as to how subpopulations shape during development and how they possibly interact with local microenvironments, as suggested in Reviewer 2, concern 1. One way of resolving this concern is to add discussions about recent relevant studies, as indicated in the reviewers' comments. And if possible, related experiments can be performed to sufficiently address the concern.

The key message of this paper is the identification of subpopulation plasmatocytes that exhibit functionally distinct behaviours compared to the overall complement of blood cells. As such, our work moves the field beyond description of largely uncharacterised sub-groups in larval blood found via scRNAseq, data that emerged during preparation of our manuscript.

We look forward to investigating whether subpopulations home to specific tissues or are specified in situ but this is significantly beyond the scope of this work as it requires long-term live imaging of larvae or pupae. We look forward to exploring this in more detail in the future. We have however discussed existing precedents in the discussion from page 22, line 511.

7. Related to point 6, recent scRNA-seq studies proposed subtypes/subpopulations of hemocytes based on differential transcriptome profiles. The authors can suggest and correlate the identified subtypes with previous analyses in the discussion (as suggested by Reviewer 2, concern 1).

We now include a paragraph in the discussion in which we suggest how the subpopulations we have uncovered may relate to the clusters that have been identified via scRNAseq approaches. This appears on page 20, line 466.

8. As pointed in Reviewer 2 concern 4, srp-gal4 is relatively weak in larval hemocytes and may not be the most appropriate enhancer to use in the stage. It seems it is too late to repeat every experiment with an alternative driver. However, I would suggest to sufficiently discuss the issue in the discussion.

This is a fair point; however the reviewer is correct in pointing out that it is too late to repeat experiments with another driver. However, we would suggest that the *srp* driver is appropriate (or at least sufficient) in this context. Other groups have successfully used the same promoter fragment to label larval hemocytes (Gyoergy et al., 2018) and our positive controls (*srp-AD* with *srp-DBD*) show clear labelling of large numbers of larval hemocytes validating the use of *srp-AD* in concert with *VT-DBD* at these stages (Figure 4b; new Video 3). This is described from page 9, line 198.

However, to address this criticism more directly we have also dissected cells from larvae in which the original *VT-GAL4* constructs were used to drive expression from *UAS-stinger* and quantified the proportions labelled (Figure 4—figure supplement 1A). This approach did not label more cells than the split GAL4 approach and, if anything, the split GAL4 approach is more sensitive. This suggests the split GAL4 approach’s reliance on *serpent*-based drivers does not dramatically alter the result and that *serpent*-negative, VT-positive cells do not make a large contribution to these subpopulations.

9. This is an optional point. Two of the reviewers suggested reconsidering the title.

We have removed “regulated by efferocytosis” from the title. The title now reflects the most important aspect of the paper – the discovery of functional differences that can be associated with specific and discrete cell populations.

Reviewer #1:Evans and colleagues used VT enhancer reporter lines to label plasmatocyte subpopulations and observed distinct subpopulations that are varied over development. Then the authors take on to characterize differential functions of these cells and found that each subpopulation exhibits variable wound responses, migratory activities, and reactivity to cell death. Further, the authors showed that overexpression of cnx14D, which is coupled with one of the VT-gal4, improves wound responses similar to the cnx14D-associated enhancer.This timely study suggests that hemocytes are more heterogeneous than has been thought. Moreover, functional analyses highlight that each subpopulation may exhibit differential activities to immune/wound responses. Related to recent single-cell studies where different subtypes of hemocytes are shown, this study will provide functional insights into how each plasmatocyte subpopulation behaves differently under diverse responses. Overall, I found this study interesting and will be of broad interest to the immunity/hematopoiesis field, however, there are several points that need to be substantiated with regard to the characterization of subpopulations and functional analyses of cnx14D.1. Characterization of the subpopulationa. In the recent scRNA seq analyses, the subpopulation/subtype/subclass of hemocytes was defined by differential expressions of multiple genes or by transcriptional activities. In the other study, subtypes of lamellocytes derived upon wasp infestation were validated by multiple reporter lines as well as protein expressions (Anderl et al., 2016). Is the expression of VT enhancer alone sufficient for defining the subpopulation? Co-expression of VT+ subpopulations with other markers and quantitation of each population would clarify and strengthen the main findings of this work.

We have addressed this as part of our response to essential revision 2 with new experimental data.

b. The authors showed that VT+ subpopulations change their numbers during the embryonic development, and most of the subpopulations are significantly reduced or even absent in the 3rd instar larvae. These patterns can be explained by changes in the enhancer activity and may not reflect the loss of specific cells. Are hemocytes that had turned on a VT-gal4 remain afterward (larval to WPP) while shutting off its expression? Or do they disappear or die? This question can be in part addressed by the tracing experiment (UAS-Gtrace).

We have addressed these points as part of our response to essential revision 1, quantifying L1 and L2 larvae and performing lineage tracing as suggested.

c. We can predict possible target genes of each VT enhancer by their genomic locations. However, enhancer activity does not always reflect the expressions of the associated gene. The authors showed the functional correlation of VT62766 cells with cnx14D without providing evidence for cnx14D expression in the VT62766 subpopulation. Is cnx14D found in the recent scRNA seq data? If so, how does it correlate to (number, developmental stage) the pattern shown by VT62766? If not, can it be validated by mRNA FISH?

We have addressed these points as part of our response to essential revisions 3 and 4.

2. The authors showed that overexpression of cnx14D in most hemocytes enhances wound responses with more plasmatocytes responding. This phenotype could be due to altered Ca^2+^ dynamics given the function of Ca^2+^ in the wound response as the authors mentioned and does not sufficiently prove that enhanced wound response seen in VT62766+ cells is caused by cnx14D. Does cnx14D RNAi reduce the wound response of VT62766+ hemocytes? And how do the authors expect the 3' end enhancer function in the cnx14D expression or function?

We thank the reviewer for these comments and agree understanding how Cnx14D impacts calcium dynamics is a fascinating new area of research – one that is being actively pursued in the Evans and Zeidler labs at this time. It is however significantly beyond the scope of this paper, but will be the subject and focus of a further publication in due course.

Regarding the role of the *VT62766* enhancer, which sits 3’ to *Cnx14D* – it is well established that enhancers can operate at either ends of a gene to regulate its transcription. For example, the enhancers that regulate *even skipped* expression for stripes 4+6 and 1+5 lie 3’ to that gene (Borok et al., 2010 Development).

3. Hemocyte populations in larvae and adults need better quantitation and characterization (Figure 3-5). As the whole larvae image only displays the sessile portion of hemocytes, it should be accompanied by proper bleeding of hemocytes. Bleeding hemocytes before and after vortexing, in case of larvae, will separate sessile and circulating hemocyte pools and quantitation of each VT+ cells per total DAPI or other hemocyte markers will provide a clear overview of each subpopulation. Similarly, adult hemocytes can be quantified after bleeding. Moreover, hemocytes attached to each body part can be quantified as shown in Bosch et al. 2019. Since the authors claim that each subpopulation is closely associated with specific sites, this should be quantified.

These comments have been addressed in response to essential revisions 3-6.

Reviewer #2:In the model organism *Drosophila*, Coates et al. identify functionally distinct macrophage (plasmatocyte) populations. Findings will inform *Drosophila* hematopoiesis, immunity, and wound healing. The authors label macrophage subpopulations through enhancer element GAL4 drivers, which show dynamic patterns of expression during development and in the ageing adult fly. Testing by criteria such as wound responses, and phagocytosis of apoptotic cells, the authors find differential engagement of certain macrophage subpopulations. They further find that reporter expression in subpopulations changes during development and shifts under conditions such as enhanced organismal apoptosis. Lastly, the authors provide proof of principle how this enhancer collection can lead to gene discovery: they identify calnexin 14D as a gene that promotes wound responses.Overall this study is based on an interesting approach. Experiments are carefully conducted. However, in the current state the study misses more wide-reaching conclusions. In some cases the authors seem to make rather unlikely speculations, which should be tested in order to draw solid conclusions. Also, existing literature should be taken more into consideration and cited accordingly.

We thank the reviewer for their comments and hope that the extra experiments we have conducted satisfactorily address their points.

1. While the authors have surveyed the enhancer-based lines quite broadly, the manuscript does not convey a clear take-home message. What do we really learn from this study functionally? For example, are hemocyte populations shaped by their environment and function, or vice versa do they home to their anatomical sites driven by their expression makeup? While the authors seem to gravitate toward the latter explanation, I think the former is much more likely, also because there is precedent in vertebrates: Lavin et al. Cell 2014, and Gosselin et al. Cell 2014 showed that tissue-resident macrophage enhancer landscapes, and tissue-specific macrophage identities, are shaped by local microenvironments. I think it would be important to investigate this question in more detail, so the authors can draw conclusions based on their findings, whichever way it may be.

The main message of this paper is the identification of functionally-distinct subpopulations of plasmatocytes. This links molecular differences to specific and distinct cellular behaviours. How subpopulations are specified by local microenvironments is an interesting (but supplementary and secondary) question that would not be conceivable without having first defined subpopulations. Nonetheless, this work identifies at least one mechanism (apoptotic cell clearance) regulating the identities we have found. We have added the existing precedents on page 22, line 511, but believe that investigating this in more detail lies beyond the scope of this paper.

2. During development and in the adult, the authors find that certain reporters cease to be detected; likely these hemocyte subpopulations are not lost but the reporter expression is simply downregulated. However, this needs to be properly analyzed by some kind of lineage tracing, such as GTRACE or flipout-lacZ lineage tracing.

See essential revision point 1.

3. Can the authors, based on their findings of functionally distinct hemocyte behaviors, suggest correlation of certain subgroups of this study with differentially expressing clusters in scRNAseq studies that were recently published? How about plasmatocyte clusters that may resemble vertebrate M1 or M2 macrophages?

These questions have been addressed in response to essential revision 7.

4. Srp-GAL4 is not the best driver for hemocytes in the larva, and may be weak in many plasmatocytes by itself. With this approach the authors may overlook many cells that form plasmatocyte subpopulations. I think it would make more sense to perform split GAL4 expression with one part of GAL4 expressed under Hml∆ control.

This has been addressed in response to essential revision 8.

5. In the adult, do the expression domains really overlap with the general srp expression? It doesn't look like that to me in some of the current images, and it would be helpful if this could be shown in closeups, and highlighted in the images. Also, some kind of model that would indicate the different populations would be helpful.

These comments have been addressed in response to essential revisions 3-5.

6. The authors restrict quantification of hemocyte subsets to the population of hemocytes in the VNC. Would the relative fraction of hemocytes be different elsewhere?

A new figure shows these additional quantifications (Figure 2—figure supplement 1b-g) and is also covered in essential revision point 3. Quantification of L1, L3 and adult subpopulations are also now provided. See previous responses.

7. Late 3rd instar larvae were used to release hemocytes. However since at that point hemocytes already enter circulation, their defined expression properties may start to be lost, if indeed hemocyte subgroups are defined by factors from local microenvironments. Perhaps it would be beneficial to look earlier (second or early third instar), when hemocytes are still more resident?

We have now characterised L1 and L2 larvae (Figure 3); see also essential revision point 4.

Reviewer #3:This is a solid study, which reveals that *Drosophila* plasmatocytes (macrophages) are not a uniform population as it was considered previously. The authors identified and characterized distinct subpopulations of the fly macrophages, which exhibit molecular and functional differences, and specific localization during different developmental stages. This exciting finding establishes heterogeneity in the innate immune system, which is not connected to adaptive immunity. The authors also reveal that subpopulation sizes are affected by increased apoptosis, demonstrating plasticity of the system. In addition, genes associated with the subpopulation-specific enhancers might act as the effector genes of the specific function, which is nicely exemplified with overexpression of calnexin14-D and opens new directions in clarifying the mechanisms regulating macrophage diversity.

We thank the reviewer for their very positive comments.

I am reluctant to ask for additional experiments but it would be interesting to test whether other known genes involved in efferocytosis by macrophages like crq show differential expression in distinct subpopulations. It could be examined by immunostaining of different VT-Gal4>GFP labeled populations with anti-Crq, anti-Draper and anti-SIMU antibodies.

This has been investigated using enhancer-based reporters (see response to essential revision 2).